# Influence of relay intercropping of barley with chickpea on biochemical characteristics and yield under water stress

Negin Mohavieh Assadi[1], Ehsan Bijanzadeh[2]*

1 Former Graduate Student of Agroecology Department, College of Agriculture and Natural Resources of Darab, Shiraz University, Shiraz, Iran, 2 Associate Professor of Agroecology Department, College of Agriculture and Natural Resources of Darab, Shiraz University, Shiraz, Iran

* bijanzd@shirazu.ac.ir

## Abstract

Relay intercropping of legumes with cereals is a useful technique for yield increment. Intercropping may affect the photosynthetic pigments, enzyme activity and yield of barley and chickpea under water stress. To investigate the effect of relay intercropping of barley with chickpea on pigment content, enzyme activity and yield under water stress, a field experiment was conducted during 2017 and 2018. The treatments included irrigation regimes (normal irrigation and cutting off irrigation at the milk development stage) as the main plot. Also, cropping systems as sub plot consisted of sole and relay intercropping of barley with chickpea in two sowing dates (December vs January). Under water stress, the early establishment of barley in December intercropped with chickpea in January ($b_1c_2$) enhanced the leaf chlorophyll content by 16% compared to sole cropping due to less competition with chickpea. Late sowing of chickpea enhanced the leaf carotenoid content of chickpea, catalase and peroxidase activities. Barley-chickpea intercropping enhanced the WUE and guaranteed a more efficient use of space (land equivalent ratio of more than 1) compared with sole crops. Under water stress, in $b_1c_2$ enhancement of total chlorophyll and water use efficiency caused to increase the grain yield of barley. In $b_1c_2$, barley and chickpea reacted to water stress with increasing total chlorophyll and enzyme activity, respectively. In this relay intercropping treatment, each crop occupied and used the growth resources from different ecological niches at different times, which is recommended in semi-arid areas.

## Introduction

An intercropping system is growing two or more plants at the same time and area [1–3]. In comparison to monoculture, intercropping mainly enhances the crop yield by more effective utilization of water resources and land [4, 5]. Relay intercropping describes a cropping system in which the life cycle of one crop overlaps that of another crop [2]. Relay intercropping of legumes with cereals is a common practice in many regions due to its economic profitability, pest and weed suppression, high productivity and environmental protection [6]. Also, the

Grant Agreement No. 97GRSM1886. The Shiraz University had no role in study design, data collection and analysis, decision to publish, or preparation of the manuscript.

**Competing interests:** The authors declare no conflict of interest

relay intercropping of cereal-legume is widely practiced in areas where the growing season is very short for two crops and rainfall declined during reproductive growth stages [7–9]. On the other hand, the relay intercropping system is used more by smallholder farmers and traditional agriculture in dry and tropical areas. In this system, most of the plants are often cultivated manually [2, 10].

Barley (*Hordeum vulgare* L.) is the second main cereal and chickpea (*Cicer arietinum* L.) as third most important legume plays an important role in agriculture of the world [3, 7]. After India, Turkey and Pakistan, Iran is the fourth biggest chickpea producer [11]. For small farmers, barley intercropped with chickpea is one of the suitable kinds of intercropping systems in the cool season of semi-arid areas [12–14]. In arid and semi-arid areas, water stress is one of the main limiting factors for crop production due to reduce water and nutrient uptake [15, 16]. The important challenge in intercropping is increasing the crop production by less water consumption [5]. Cereal-legume intercropping is an advantageous intercropping system under water stress [16]. However, crop species in intercropping may differ in their responses to growth under water shortage [5, 16].

Photosynthetic pigments are extraordinarily sensitive to water stress which are the main indicator of water deficit [17, 18]. Water stress accelerates the leaf senescence of crops and consequently chlorophyll content and photosynthesis rates are reduced, negatively [19]. In *Arabidopsis thaliana*, chlorophyll content, chlorophyll fluorescence and RWC declined gradually under water stress [20]. It has been declared that under water stress, the total chlorophyll content decreased because of degradation of chlorophyll *a* [18, 21]. Intercropping systems enhance the chlorophyll content in leaves by improving the nitrogen availability for plants [22]. In peppermint intercropped with soybean, chlorophyll content increased 17 to 27% compared to peppermint monoculture, which correlated to yield enhancement, positively [23]. Also, in cropping systems, to diminish the negative effect of water shortage on growth rate, plants have some defense mechanisms such as enhanced antioxidant activities [24, 25]. Crops usually enhance the activity of catalase (CAT) and peroxidase (POD) in response to water stress [26, 27]. Water stress triggers leaf water loss, which decreases relative water content (RWC), and limits growth rate, consequently [28, 29].

In a common intercropping, the yield of each crop is usually less than sole crop, while summation of relative yields is often higher than one [30, 31]. In contrast, some studies state that intercropping creates a significant yield advantage compared to monoculture [1, 32]. Overall, the yield of an intercropping system is related to many factors such as sowing date, plant density, crop type, competition ratio and biotic and abiotic stress levels [10, 33, 34]. In a field study, barley-chickpea intercropping improved the sum yield in comparison to sole barley [14]. In fact, the interspecific interaction enhances the nutrient and water absorption from different depths of soil profile, which caused yield enhancement [35]. Rahimi Azar et al. [36] suggested that the highest yield of chickpea was obtained from intercropping chickpea with barley as a 1:1 ratio. In contrast, the chickpea seed yield was influenced by intercropping with barley, negatively [37].

Unfortunately, in some years, there is no considerable rainfall in March to May when the water requirement for crops increases to complete the seed filling period [38]. These conditions are usually typical in the south of Iran, which has a dry and hot spring. With respect to the occurrence of the rainfall in the cool season, the farmers have to irrigate the crop after anthesis [29]. One strategy to increase grain yield production of crops can apply the relay intercropping to withstand environmental stresses [39]. In relay intercropping, crop type and the sowing date of each component are crucial to the final yield. At the best of our knowledge, no experiment has been carried out to consider the influence of relay intercropping of barley with chickpea on biochemical properties of each crop under water stress. It's hypothesized that

**Table 1. Some physical and chemical characteristics of the soil in the experimental site (depth of 0–30 cm) during 2017 and 2018 growing seasons.** P content of the soil is Olsen P and N content of the soil was determined by the Kjeldhal method.

| Year | Sand (%) | Silt (%) | Clay (%) | SOM (%) | Electrical conductivity (dS/m) | pH | N (mg/kg) | P (mg/kg) | K (mg/kg) |
|------|----------|----------|----------|---------|-------------------------------|-----|-----------|-----------|-----------|
| **2017** | 38.01 | 43.95 | 18.04 | 0.821 | 1.011 | 7.31 | 0.75 | 51 | 311 |
| **2018** | 38.12 | 44.21 | 17.67 | 0.843 | 1.015 | 7.33 | 0.74 | 52 | 307 |

relay intercropping influenced the photosynthetic pigment, antioxidant activities and yield of barley and chickpea after cutting off irrigation in the late season. It's also believed that the crops that have more photosynthetic pigments and enzyme activity in the relay intercropping of barley and chickpea can produce more grain yield under water stress conditions. The main goal of this field experiment was to detect the suitable relay intercropping system of barley-chickpea under water stress conditions and evaluate the biochemical changes of each crop in intercropping.

## Material and methods

### Filed experiment description and treatments

A consecutive two-year field experiment was carried out to evaluate the effect of late season water stress and different combinations of relay intercropping of chickpea with barley on some biochemical traits and yields. The experiment was conducted at College of Agriculture and Natural Resources of Darab (28°29′ N, 54°55′ E), Darab, Fars province, Iran during the 2017 and 2018 growing seasons. The soil type in the experiment site was loam (fine, loamy, carbonatic, hyperthermic, typic Torriorthents) [40] and the other soil characteristics are given in Table 1.

### Weather conditions and total water applied

The study area has a semi-arid climate with cool and rainy winters and dry and hot summers. Likewise, weather data for the research field during both of the years are presented in Table 2. During the growth period of barley and chickpea from December to May, the mean temperature in the 2018 growing season was more than 2017 (Table 2). In each year, the maximum temperatures were registered in April-May and the minimum in December- January. By increasing the mean temperature, the water evaporation in the second year (949.7 mm) increased compared with the first year (853.6 mm). Also, total rainfall during the active growth stages of 2017 and 2018 was 240.2 and 164.1 mm, respectively. A significant portion of rainfall occurred in January (94.2 mm) and March (99.5 mm) of the first year (Table 2). On the other

**Table 2. Minimum and maximum air temperatures, monthly rainfall, and pan evaporation of the experimental site during 2017 and 2018 growing seasons.**

| Month | Temperature (ºC) | | | | | | Rainfall (mm) | | Pan evaporation (mm) | |
|-------|-----|-----|------|-----|-----|------|---------|---------|---------|---------|
| | 2017–18 | | | 2018–19 | | | 2017–18 | 2018–19 | 2017–18 | 2018–19 |
| | Min | Max | Mean | Min | Max | Mean | | | | |
| December | 5.1 | 20.1 | 12.6 | 5.7 | 21.2 | 13.5 | 2.2 | 24.8 | 77.6 | 88.4 |
| January | 4.4 | 19.2 | 11.8 | 3.8 | 23.4 | 13.6 | 94.2 | 78.3 | 66.3 | 78.5 |
| February | 8.6 | 21.4 | 15.0 | 9.2 | 23.6 | 17.9 | 44.3 | 38.9 | 88.1 | 91.2 |
| March | 10.5 | 25.4 | 18.0 | 8.1 | 28.5 | 19.3 | 99.5 | 22.1 | 143.1 | 175.3 |
| April | 12.9 | 29.8 | 21.4 | 8.9 | 33.2 | 22.6 | 0.0 | 0.0 | 202.3 | 218.9 |
| May | 17.9 | 34.4 | 26.2 | 15.8 | 38.1 | 30.0 | 0.0 | 0.0 | 276.2 | 297.4 |
| Total | | | | | | | 240.2 | 164.1 | 853.6 | 949.7 |

hand, in two years, there was no efficient rainfall in April to May when the water demand for barley and chickpea enhanced sharply to complete the grain filling stage (Table 2). During the 2017 and 2018 growing seasons, in normal irrigation and water stress regimes, intercropping of barley with chickpea in $b_1c_2$, $b_2c_1$, $b_2c_2$ treatment consumed less water (irrigation amount + rainfall) compared to sole cropping, significantly ($p \leq 0.05$) (Fig 1A and 1B). In $b_1c_1$, early sowing of barley and chickpea at the same time in December, increased the competition between two crops for water uptake. In the second year, barley and chickpea consumed more water (Fig 1B) due to more evaporation and higher mean temperatures, especially in April to May (Table 2).

Each year the experiment was conducted as split plot based on a randomized complete block design with three replications. The treatments included irrigation regimes at two levels (normal irrigation and cutting off irrigation at the milk development stage of barley [Zadoks growth stage (ZGS70)] [41] were as the main plot. Cropping system treatments as sub plot consisted of sole cropping of barley in December ($b_1$) and January ($b_2$), sole cropping of chickpea cultivar in December ($c_1$) and January ($c_2$), and different combinations of intercropping consisted of intercropping of barley + chickpea in December ($b_1c_1$), intercropping of barley in December + chickpea in January ($b_1c_2$), intercropping of barley in January + chickpea in December ($b_2c_1$) and intercropping of barley + chickpea in January ($b_2c_2$). The seeds of barley (Zehak cultivar) and chickpea (Darab cultivar) were provided from the Agriculture and Natural Resources Research Center of Darab, Fars Province, Iran. Zehak is a six-rowed barley cultivar with a medium plant height of 50–80 cm. The growing season length for Zehak is approximately 150 days, which is specifically adapted to growing in the warm and arid regions of Iran [42]. Also, Darab is an early mature cultivar of chickpea with a plant height of 28 cm, and has a semi-upright position, which is suitable for arid/semi-arid conditions [43].

The plot size was 3m×2m and it was surrounded with a 30 cm high earth berm by a 1m wide buffer space between the plots. The seedbed was prepared by mouldboard ploughing and disking. The plowing dept was 30 cm. The uniform seeds of barley and chickpea were sown handily at a soil depth of 3 cm, giving 250 and 40 plants $m^{-2}$ planting density, respectively. The barley and chickpea were designed as replacement series with a ratio of one row of barley and one row of chickpea in intercropping plots, where the inter-row space between barley and chickpea was 30 cm. Based on soil test (Table 1), phosphorus (P) as superphosphate triple source at the rate of 50 kg $ha^{-1}$ and Nitrogen (N) as urea source at the rate of 60 kg $ha^{-1}$ were used in the field experiment. Total P and half dose of N used in the soil at sowing and remaining half N dose used by irrigation water at beginning of the stem elongation stage of barley (ZGS31). The seeds were sown on December 15th or January 15th, based on sowing dates of cropping systems.

The effective root zone is the depth within which most crop roots are concentrated, which was estimated as ∼50–100 cm for barley and as ∼60–70 cm for chickpea [44]. Thus, with respect to monitoring the soil water content at the root zone, the soil water content was traced in each plot from 30 cm depth down to 90 cm, gravimetrically. Before each irrigation, the soil profile was sampled up to 90-cm by an auger. Then, the volume of water applied in normal irrigation was accorded to restoring root zone moisture deficit (when 50% of available water was depleted in effective root-zone depth of 90 cm) to near-field capacity [29]. A surface drip irrigation system was applied for irrigation. A 20 mm diameter polyethylene pipe with in-line drippers at 40 cm intervals was placed on one side of each planting row. Overall, plots were irrigated five times for normal irrigation and three times for cutting off irrigation at the milk development stage of barley (ZGS70) of barley. Total water applied ($m^3$) (irrigation amount + rainfall) in each irrigation regime and cropping system in 2017 and 2018 growing seasons are presented in Fig 1.

**(a)**

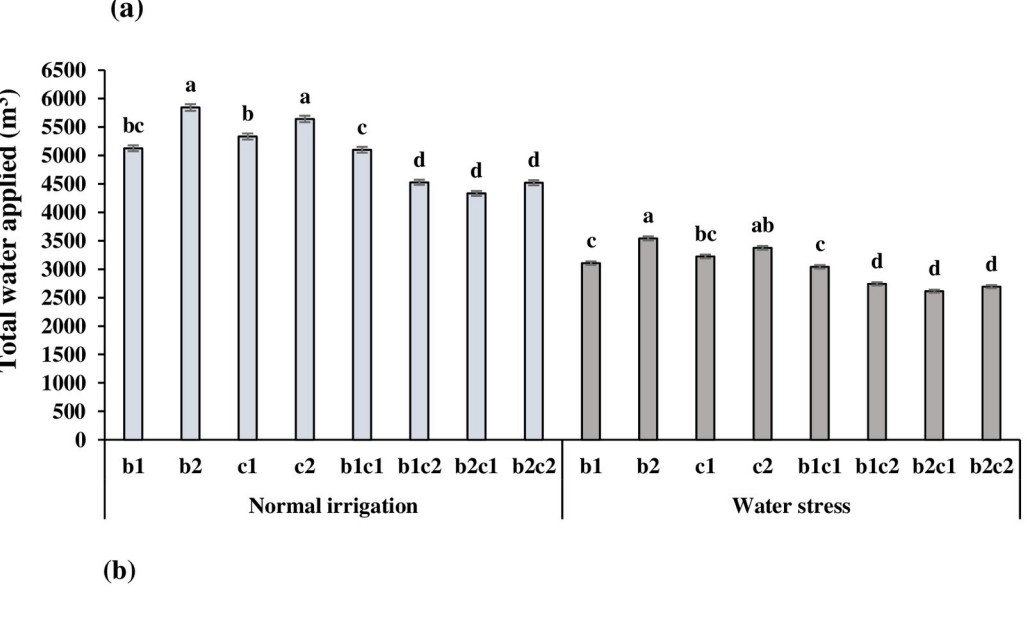

**(b)**

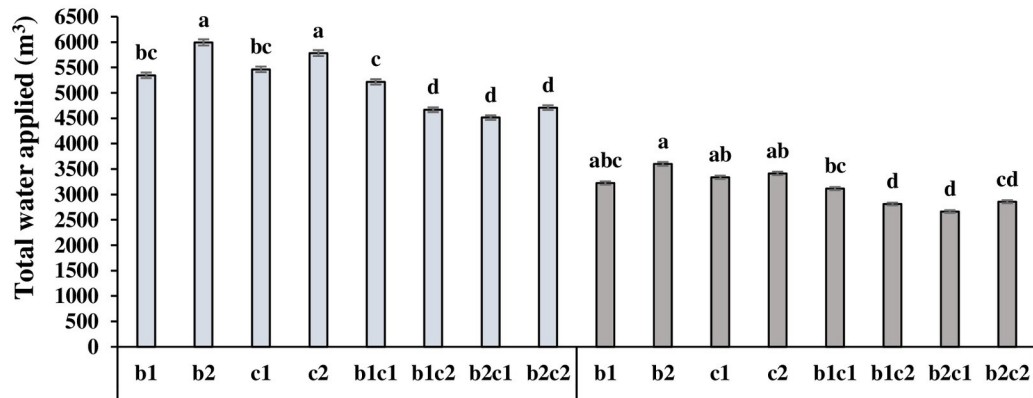

**Fig 1.** Total water applied ($m^3$) in each irrigation regime and cropping system during the 2017 (a) and 2018 (b) growing seasons. $b_1$: Sole cropping of barley in December $b_2$: Sole cropping of barley in January, $c_1$: Sole cropping of chickpea in December $c_2$: Sole cropping of chickpea in January, $b_1c_1$: Intercropping of barley + chickpea in December; $b_1c_2$: Intercropping of barley in December + chickpea in January, $b_2c_1$: Intercropping of barley in January + chickpea in December, $b_2c_2$: Intercropping of barley + chickpea in January. In each irrigation regime, values with a letter in common should not be considered different at 0.05 probability based on Tukey-Kramer test. Bars represent mean ± SE.

To determine the photosynthetic pigments, antioxidant enzyme activity and RWC, the top leaf of barley and chickpea were sampled at the end of the milk development stage of barley (ZGS 77). At the crop maturity stage in May, the plants in the central 1 $m^2$ of each plot were hand harvested. Then, the samples were oven dried at 72°C for 48 h and weighted for biological yield. Finally, the samples were threshed and grains were separated and weighed for grain yield determination.

## Chlorophyll and carotenoid content assessment

The chlorophyll content was measured by fresh tissue of the top leaf in each plot. Ten ml of 80% acetone was added to 200 mg of leaf tissue gradually and ground by a mortar and pestle. The created slurry was centrifuged for 10 min at 4000 rpm, and the supernatant was filtered

through Whatman No. 2 filter paper placed in a funnel as the solution was transferred. Absorbance was measured by a double-beam UV-VIS spectrophotometer (UV-1900 spectrophotometer, Shimadzu, Japan) at $\lambda$ = 645, 663, and 470 nm. Chlorophyll *a*, *b* and total and Carotenoid were calculated according to the following equations [45]:

$$\text{Chlorophyll } a = (19.3 \times \text{A663} - 0.86 \times \text{A645}) \text{ V}/100 \text{ W} \tag{1}$$

$$\text{Chlorophyll } b = (19.3 \times \text{A645} - 3.6 \times \text{A663}) \text{ V}/100 \text{ W} \tag{2}$$

$$\text{Total Chlorophyll } = \text{Chlorophyll } a + \text{Chlorophyll } b \tag{3}$$

$$\text{Carotenoid } = (1000 \text{ A470} - 1.82 \text{ Ch. } a - 85.02 \text{ Ch. } b) \tag{4}$$

Where, V is volume of purified solution, W is leaf fresh weight, and A663, A645, and A470 were optical absorption wavelengths at 663, 645, and 470 nm, respectively.

## Antioxidant enzymes assay

For enzyme extraction, 0.5 g of fresh leaves were ground to fine powder in liquid nitrogen by mortar and pestle and then homogenized in 2 mL extraction buffer containing 10% (w/v) polyvinyl pyrrolidone (PVP) in 50 mM potassium-phosphate buffer (pH 8), 1 mM dithiothreitol (DTT), and 0.1 mM ethylene diamine tetra acetic acid (EDTA). The homogenate was centrifuged at 20,000×g (4°C) for 30 minutes. The supernatant was used to assess antioxidant enzymes of catalase and peroxidase.

## Catalase enzyme

The catalase enzyme activity (CAT) was determined using spectrophotometer (UV-160A) according to the method of Aebi [46], by monitoring the decrease in absorbance at 240 nm because of $H_2O_2$ consumption. One mL of reaction mixture contained 50 mM potassium phosphate buffer (pH = 7.0) and 15 mM $H_2O_2$. The reaction was initiated by adding 50 μL of crude extract to this solution. CAT activity was expressed as units (μmol $H_2O_2$ consumed per minute) per milligram of protein.

## Peroxidase enzyme

The peroxidase enzyme activity (POD) was evaluated by the method of Chance and Maehly [47]. One mL of reaction mixture contained 13 mM guaiacol, 50 mM potassium phosphate buffer (pH 7) and 5 mM $H_2O_2$. An increase in absorbance because of oxidation of guaiacol (extinction coefficient: 26.6 $mM^{-1}.cm^{-1}$) was traced at 470 nm for a minute. Peroxidase activity was expressed as units (μmol guaiacol oxidized per minute) per milligram of protein.

## Leaf relative water content

The leaf relative water content (RWC) was measured by method of Machado and Paulsen [48]. Eight leaf discs (8 mm in diameter) from the fully expanded flag leaf were weighed for determination of fresh weight (FW). The leaf discs were kept in distilled water for 6 h, then dried with filter paper and weighed for determination of total weight (TW). Dry weight (DW) was determined after drying the discs at 70°C in an oven for 24 h. Finally, the RWC was determined as:

$$\text{RWC} = [(\text{FW} - \text{DW})/(\text{TW} - \text{DW})] \times 100$$

## Water use efficiency

The water use efficiency (WUE) in each treatment was evaluated as the ratio of seed yield (g. $m^{-2}$) to total water consumed (mm) [49].

## Land equivalent ratio

The land equivalent ratio (LER) is expressed as the land equivalent required for growing either crop in intercropping compared to the land area required for sole cropping of each crop. The LER total ($LER_t$) was calculated as [50]:

$$LERt = LERb + LERc$$

$$LERb = \frac{Ybi}{Ybm} \tag{5}$$

$$LERc = \frac{Yci}{Ycm} \tag{6}$$

Where $LER_b$ and $LER_c$ were land equivalent ratios of barley and chickpea, respectively; $Y_{bi}$ m and $Y_{ci}$ were the yields of barley and chickpea in monoculture; and Ybi and Yci were the yields of barley and chickpea in the intercropping system. When the $LER_t$ value was more than one, intercropping was more useful compared to sole cropping. Controversy, when the LER was less than one, intercropping affected the yield of crops, negatively [6].

## Competition ratio

The competition ratio (CR) is a suitable index to determine the competitive ability between two crops in intercropping. CR shows stronger competitive ability to the species and is more beneficial compared to other indices. The CR was determined by the following equations [6]:

$$CR = CR_b + CR_c \tag{7}$$

$$CRb = \frac{(LERb)}{(LERc)} \times \frac{(Zci)}{(Zbi)} \tag{8}$$

$$CRc = \frac{(LERb)}{(LERc)} \times \frac{(Zci)}{(Zbi)} \tag{9}$$

Where $Z_{bi}$ is the sown proportion of barley intercropped with chickpea and $Z_{ci}$ is the sown proportion of chickpea intercropped with barley. $CR_b$ is the competition ratio of barley and $CR_c$ is the competition ratio of chickpea.

## Statistical analyses

The model used for variables in analysis of variance was fix model for all variables including year, irrigation regime and cropping system. In order to check the normality distribution of data, Kolmogorov-Smirnov and Shapiro-Wilk tests were used and the skewness and kurtosis indices of data proved that the distribution of data was normal. The F-test was carried out to check the equality of variances. The residuals of the model were also normal using q-q plot. Data were analyzed by SAS software 2012 (version 9.4) and the means were compared by Tukey-Kramer test at 0.05 probability level ($p \leq 0.05$). Because of the significant effect of the

**Table 3. Combined analysis of variance for total water applied, pigment content and enzyme activity of barley and chickpea.** Irrigation regimes (I) included normal irrigation and cutting off irrigation at the milk development stage of barley and cropping system (c) consisted of sole and relay intercropping of barley and chickpea in December and January.

| Source of variance (S.O.V) | Degrees of freedom (df) | Mean squares (MS) | | | | | | | | | | | | | |
|---|---|---|---|---|---|---|---|---|---|---|---|---|---|---|---|
| | | TWA | | Chl. $a$ | | Chl. $b$ | | CAR | | Total Chl. | | CAT | | POX | |
| | | Barley | Chickpea | Barley | Chickpea | Barley | Chickpea | Barley | Chickpea | Barley | Chickpea | Barley | Chickpea | Barley | Chickpea |
| Year(Y) | 1 | 12.5** | 10.33* | 0.76* | 1.21* | 0.13* | 1.56* | 3.96 * | 14.56* | 5.32* | 11.96* | 5.86* | 4.63* | 5.36* | 2.36* |
| Y× Replication | 4 | 0.025ns | 0.066ns | 0.0088ns | 0.086 ns | 0.0002 ns | 0.0063ns | 0.0031 ns | 0.036 ns | 0.0061 ns | 0.065 ns | 0.0073 ns | 0.0036 ns | 0.086 ns | 0.031 ns |
| Irrigation regime (I) | 1 | 7.65* | 11.26* | 0.027* | 0.056* | 0.0178* | 0.189* | 0.014ns | 0.086 ns | 0.167* | 0.986* | 0.351* | 0.236* | 0.498* | 0.569ns |
| Cropping system (C) | 5 | 4.36* | 3.68* | 0.0921** | 0.653* | 0.0309** | 0.410* | 0.0013ns | 0.013* | 0.052* | 0.056* | 0.098ns | 0.088* | 0.103* | 0.201ns |
| I×C | 5 | 11.57* | 14.63 ns | 0.00768 ns | 0.00563* | 0.0023* | 0.036 ns | 0.0031* | 0.036 ns | 0.0136* | 0.0189* | 0.0436 ns | 0.0365 ns | 0.891 ns | 0.0986* |
| Y×I | 1 | 23.45 | 12.98* | 0.00019 ns | 0.00086 | 0.000046 ns | 0.0021 | 0.00018ns | 0.0025 | 0.00065 ns | 0.0658 | 0.00087 | 0.0035 | 0.0023 | 0.0064 |
| Y×C | 5 | 2657.3 | 5684.3* | 0.000023ns | 0.000043 | 0.000021 ns | 0.00613 | 0.00015 ns | 0.0014 | 0.00046 ns | 0.045 | 0.000086 | 0.000011 | 0.00095 | 0.00068 |
| Y×I×C | 5 | 456879.3* | 436357.3* | 0.00061* | 0.00096* | 0.0003* | 0.0032* | 0.00031* | 0.00894* | 0.00025* | 0.00035* | 0.00032* | 0.00057* | 0.0092* | 0.0035* |
| Error | 35 | 10.33 | 9.66 | 0.002433 | 0.002369 | 0.002136 | 0.03561 | 0.00463 | 0.0261 | 0.0007862 | 0.008661 | 0.009823 | 0.006357 | 0.01235 | 0.01276 |
| CV% | | 4.87 | 11.23 | 4.68 | 5.39 | 5.23 | 4.98 | 3.36 | 11.69 | 6.11 | 12.37 | 5.11 | 3.98 | 6.54 | 7.93 |

TWA: total water applied; Chl. a: Chlorophyll $a$; Chl. b: Chlorophyll $b$; CAR: Carotenoid; Total chl.: Total chlorophyll; CAT: Catalase; POX: Peroxidase; LER: land equivalent ratio; CR: Competition ratio. ns, * and **: no significant and significant at the 5% and 1% probability levels, respectively.

year × irrigation regime × cropping system on considered traits, the data of two years for barley and chickpea were presented, separately.

## Results

### Analysis of variance

Results of combined analysis of variance over years demonstrated that the main effect of the year was significant on all of the considered traits at 0.05 probability level (Table 3). It might be related to different temperatures, rainfall and evaporation during the reproductive growth stages of barley and chickpea (Table 2). Also, the interaction effect of year × irrigation regime × cropping system was significant ($p \leq 0.05$) on total water applied, chlorophyll a, chlorophyll b, carotenoid, catalase and peroxidase of barley and chickpea (Table 3). On the other hand, this interaction had a significant effect at %5 probability level on relative water content, yield attributes, water use efficiency and competition indices (Table 4).

### Pigment contents

In both years, the irrigation regime and the cropping system had noticeable effects on the pigment content of barley (Table 5). In both of the irrigation regimes, early sowing of barley and chickpea in December ($b_1c_1$) in 2017, increased the chlorophyll a content of barley as well as in 2018. Under water stress, the chlorophyll a content of barley was affected by water deficit in the late season, negatively and the highest amount was obtained in $b_2$ and $b_1c_2$ treatments. The $b_1c_1$ and $b_1c_2$ intercropping treatments influenced the chlorophyll b content of barley, which is in the range of 0.39±0.007 to 0.49±0.007 mg/g FW in normal irrigation and 0.23±0.008 to 0.27±0.006 mg/g FW under water stress (Table 5). Similar to the chlorophyll a, late sowing of barley intercropped with chickpea had a significant effect on reducing the chlorophyll b content of barley (0.07±0.008 to 0.11±0.009 mg/g FW), under water stress. The highest carotenoid

**Table 4. Combined analysis of variance for relative water content, yield attributes, water use efficiency and competition indices of barley and chickpea.** Irrigation regimes (I) included normal irrigation and cutting off irrigation at the milk development stage of barley and cropping system (c) consisted of sole and relay intercropping of barley and chickpea in December and January.

| Source of variance (S.O.V) | Degrees of freedom (df) | Mean squares (MS) | | | | | | | | | | | | |
|---|---|---|---|---|---|---|---|---|---|---|---|---|---|---|
| | | RWC | | Gy | | By | | HI | | WUE | | LER | | CR | |
| | | Barley | Chickpea | Barley | Chickpea | Barley | Chickpea | Barley | Chickpea | Barley | Chickpea | Barley | Chickpea | Barley | Chickpea |
| Year(Y) | 1 | 23.44* | 12.36* | 5727.20* | 3266.3* | 6312.32* | 12386.51* | 11.31* | 10.91* | 28.93* | 98.32* | 14.33* | 9.53* | 4.32* | 2.36* |
| Y× Replication | 4 | 13.91 | 15.69 | 96.32 | 54.23 | 862.39 | 563.28 | 9.88 | 15.36 | 15.68 | 10.41 | 12.44 | 2.64 | 1.31 | 4.54 |
| Irrigation regime (I) | 1 | 655.32** | 866.34* | 6821.37* | 7433.7* | 2569.96** | 1256.38* | 154.32* | 186.76* | 986.33 | 864.38** | 5.63* | 4.36* | 2.96* | 1.76* |
| Cropping system (C) | 5 | 684.23* | 684.23* | 39154.51* | 22368.5* | 7568.64* | 6354.28* | 863.94ns | 602.31* | 765.35* | 625.97ns | 15.3* | 10.91* | 9.36* | 10.37ns |
| I×C | 5 | 244.36* | 278.39ns | 557.11* | 453.68* | 235.04* | 133.98ns | 431.22* | 45.362ns | 153.68 ns | 25.63 ns | 15.31 ns | 4.36 ns | 2.31 ns | 3.33* |
| Y×I | 1 | 11.86 ns | 50.36* | 2523.4 ns | 2438.71* | 3111.37 ns | 7423.69* | 17.44* | 10.76* | 52.93 ns | 45.36* | 11.91 ns | 12.32* | 10.69* | 9.32* |
| Y×C | 5 | 27.36 ns | 38.91* | 1589.3 ns | 1025.96 ns | 1457.56 ns | 1598.78* | 11.59* | 59.67 ns | 11.21 ns | 10.86* | 11.56* | 10.91 ns | 7.43 ns | 6.54 ns |
| Y×I×C | 5 | 5.33* | 7.36* | 22403.2* | 28456.81* | 4222.44* | 4203.56* | 8.80* | 45.36** | 43.96* | 57.48* | 22.33* | 9.86* | 5.63* | 4.36* |
| Error | 35 | 1.63 | 0.56 | 277.3 | 156.98 | 399.48 | 766.2 | 212.3 | 127.86 | 14.68 | 10.81 | 1.23 | 2.38 | 1.54 | 2.38 |
| CV% | | 7.86 | 9.35 | 10.58 | 12.36 | 11.21 | 10.43 | 12.34 | 10.41 | 8.51 | 12.29 | 4.23 | 5.11 | 6.11 | 7.57 |

RWC: relative water content; Gy: grain yield, By: biological yield; HI; harvest index; WUE: water use efficiency; LER: land equivalent ratio, CR: competition ratio. ns, * and **: no significant and significant at the 5% and 1% probability levels, respectively.

content was obtained in $b_1c_2$ under normal irrigating, while in water stress conditions, $b_2c_2$ treatment in 2017 created the maximum carotenoid content (0.41±0.008 mg/g FW), which had no significant difference (p≤ 0.05) with $b_1c_2$, $b_2c_1$ intercropping treatments. In addition, early sowing of barley ($b_1c_1$ and $b_1c_2$) had better performance in terms of total chlorophyll compared to the late sowing of barley ($b_2c_1$ and $b_2c_2$) treatments. Increasing the growing season length in early barley cultivation seems to contribute to better establishment and adaptation of barley intercropped with chickpea, so that total chlorophyll content is enhanced more than late barley cultivation. Overall, the total chlorophyll content in the 2017 was more than the 2018 growing season except for the $b_2c_2$ treatment under water stress (Table 5).

In chickpea, pigment contents were affected by the irrigation regime and the cropping system during the two years of the experiment (Table 6). Late sowing date of chickpea in January ($b_1c_2$ and $b_2c_2$) enhanced chlorophyll *a* content of chickpea compared to the other intercropping treatments. Under water stress, the chlorophyll *a* content in late sowing of chickpea with barley in the same time ($b_2c_2$) increased 25 and 29% compared to sole cropping of chickpea in January ($c_2$) in 2017 and 2018, respectively. In addition, in all of the cropping systems and irrigation regimes, the chlorophyll *a* content of chickpea in 2017 was higher than 2018 (Table 6). In normal irrigation, the highest chlorophyll *b* content of chickpea was obtained in $b_1c_1$ (0.41 ±0.003 mg/g FW) and $b_2c_2$ (0.39±0.005 mg/g FW) treatments of the first year, which was more than the second year, significantly (p≤ 0.05). A similar trend was observed under water stress conditions and the chlorophyll *b* content was in the range of 0.21±0.006 to 0.31±0.007 mg/g FW (Table 6). In 2017 and 2018, the carotenoid content of chickpea was affected by the cropping system. In both of the irrigation regimes, $b_1c_2$ and $b_2c_2$ had the higher carotenoid content compared to the other intercropping treatments in the range of 0.19±0.004 to 0.26±0.005 mg/g FW in normal irrigation and 0.17±0.005 to 0.24±0.002 mg/g FW, under water stress (Table 6). Under normal irrigation, $b_2c_1$ treatment and in water stress, $b_1c_1$ and $b_2c_1$ treatments had the lowest carotenoid content of chickpea compared to the other intercropping treatments. It

appears that late sowing of chickpea in January improved the carotenoid content of chickpea in comparison to early sowing in December. Finally, in intercropping treatments, total chlorophyll in $b_1c_2$ and $b_2c_2$ in normal irrigation and $b_2c_2$ in water stress conditions enhanced compared to early sowing of chickpea intercropped with barley, significantly ($p \leq 0.05$). Also, in both of the irrigation regimes, the total chlorophyll in the sole cropping of chickpea ($c_1$ and $c_2$) in the first year was more than the second year, significantly ($p \leq 0.05$). Under water stress, when the growing season length of chickpea decreased in $b_1c_2$ and $b_2c_2$, because of better adaptation of chickpea in the canopy of barley rows, the total chlorophyll of chickpea enhanced compared to the first sowing date (December) (Table 6).

## Antioxidant enzymes activity

The mean comparison data of catalase (CAT) and peroxidase (POX) activity of barley intercropped with chickpea under different irrigation regimes in 2017 and 2018 are presented in Table 7. In all of the cropping systems, water stress enhanced the CAT activity of barley, positively. The highest amount of CAT was created by $b_2c_1$ treatment in 2018 with no significant difference from 2017. In the second year, because of higher temperatures and evaporation during the reproductive stages (Table 2), barley might respond to water stress through enhancing the CAT activity. On the other hand, intercropping of barley with chickpea improved the activity of CAT compared to sole cropping of barley ($b_1$ and $b_2$), when crops were exposed to water stress. In normal irrigation, late sowing of barley in January ($b_2$) reduced the POX

**Table 5. Interaction effect of irrigation regime and cropping system of barley with chickpea on chlorophyll *a*, *b*, carotenoid and total chlorophyll of barley.**

| | | Barley | | | | | | | |
|---|---|---|---|---|---|---|---|---|---|
| | Cropping system | Chl. *a* (mg/g FW) | | Chl. *b* (mg/g FW) | | CAR (mg/g FW) | | Total chl. (mg/g FW) | |
| Irrigation regime | | 2017 | 2018 | 2017 | 2018 | 2017 | 2018 | 2017 | 2018 |
| Normal irrigation | $b_1$ | 1.45±0.016CDb | 1.36±0.012Db | 0.38±0.007BCbc | 0.36±0.006BCabc | 0.31±0.004BCbc | 0.33±0.005BCc | 1.83±0.011DEcd | 1.72±0.009Ec |
| | $b_2$ | 1.79±0.022ABa | 1.63±0.007BCa | 0.33±0.009BCbc | 0.30±0.004Cc | 0.28±0.003BCbc | 0.31±0.007BCc | 2.12±0.017BCb | 1.93±0.010Cb |
| | $b_1c_1$ | 1.99±0.019Aa | 1.76±0.009ABa | 0.41±0.010ABab | 0.39±0.007BCab | 0.38±0.005Bb | 0.37±0.006Bb | 2.41±0.011Aa | 2.15±0.010Ba |
| | $b_1c_2$ | 1.43±0.014CDb | 1.34±0.014Db | 0.49±0.007Aa | 0.41±0.004ABa | 0.63±0.003Aa | 0.54±0.005Aa | 1.92±0.009Cc | 1.75±0.007EFc |
| | $b_2c_1$ | 1.31±0.009Db | 1.23±0.018Db | 0.37±0.008Bbc | 0.33±0.007Bab | 0.25±0.002Cc | 0.36±0.004Bc | 1.68±0.008EFd | 1.56±0.005GHd |
| | $b_2c_2$ | 1.27±0.008Db | 1.19±0.007Db | 0.29±0.004Cc | 0.30±0.005Cc | 0.23±0.004Cc | 0.38±0.009Bc | 1.56±0.006GHe | 1.51±0.008Hd |
| Water stress | $b_1$ | 1.05±0.007BCbc | 0.79±0.008DEbc | 0.19±0.009BCabc | 0.21±0.007ABCa | 0.29±0.007BCDc | 0.23±0.007Dd | 1.24±0.005Cb | 0.99±0.007Ec |
| | $b_2$ | 1.29±0.019Aa | 1.03±0.005BCDa | 0.17±0.005CDbcd | 0.22±0.009ABCa | 0.33±0.004ABCb | 0.27±0.005CDcd | 1.46±0.006Aa | 1.25±0.006BCa |
| | $b_1c_1$ | 1.00±0.007BCcd | 0.86±0.006CDEbc | 0.23±0.008ABCab | 0.25±0.008ABa | 0.28±0.001CDc | 0.24±0.009Dd | 1.23±0.006Cb | 1.11±0.004Dbc |
| | $b_1c_2$ | 1.18±0.003ABb | 1.07±0.007ABa | 0.26±0.007Aa | 0.27±0.006Aa | 0.39±0.005Aab | 0.37±0.007ABa | 1.44±0.008Aa | 1.34±0.003Ba |
| | $b_2c_1$ | 0.74±0.003EFd | 0.56±0.011FGd | 0.11±0.009DEcd | 0.10±0.002DEb | 0.36±0.006ABCab | 0.31±0.008BCDbc | 0.85±0.004Fc | 0.66±0.005GHd |
| | $b_2c_2$ | 0.48±0.008Ge | 0.66±0.007EFGcd | 0.09±0.004Ed | 0.07±0.008Eb | 0.41±0.008Aa | 0.36±0.008ABa | 0.57±0.006Hd | 0.73±0.004Gd |

Chl. a: Chlorophyll *a*; Chl. b: Chlorophyll b; CAR: Carotenoid; Total chl.: Total chlorophyll; $b_1$: Sole cropping of barley in December; $b_2$: Sole cropping of barley in January; $b_1c_1$: Intercropping of barley + chickpea in December; $b_1c_2$: Intercropping of barley in December + chickpea in January; $b_2c_1$: Intercropping of barley in January + chickpea in December; $b_2c_2$: Intercropping of barley + chickpea in January. Means ± SE with common capital letters in each irrigation regime between two years and means ± SE with common lowercase letters in each irrigation regime and column should not be considered different at 0.05 probability based on Tukey-Kramer test.

activity of barley compared to $b_1$ treatment. In contrast, under water stress, POX activity of $b_2$ from 9.88±0.018 and 10.11±0.014 Unit/mg$^{-1}$ protein in 2017 and 2018, declined to 5.27±0.011 and 6.53±0.18 Unit/ mg protein in $b_1$, respectively (Table 7). In response to the late sowing date of barley in January to escape from the water deficit in the late season, POX activity increased in $b_2c_2$.

The CAT activity of chickpea was promoted by water stress, so that by relay intercropping of chickpea in January ($b_1c_2$) reached to 5.22±0.019 and 4.31±0.010 Unit/mg protein in 2017 and 2018, respectively (Table 8). In two irrigation regimes, late sowing of chickpea in sole cropping ($c_2$) had the lower CAT activity in the range of 0.69±0.005 to 0.94±0.009 Unit/mg protein compared to $c_1$. On the other hand, in intercropping treatments, $b_1c_1$ had the least CAT activity in range of 0.43±0.006 to 1.47±0.0015 Unit/mg protein. The POX activity of chickpea was influenced by water stress, positively and in $b_1c_2$ of normal irrigation from 2.13 ±0.017 and 2.98±0.009 Unit/ mg protein enhanced to 11.43±0.029 and 11.9±0.012 Unit/mg protein in 2017 and 2018, respectively (Table 8). Likewise, in $b_1c_2$ POX activity of chickpea increased 141 and 86% compared to $c_2$, in the 2017 and 2018 years, respectively. Under water

**Table 6. Interaction effect of irrigation regime and cropping system of barley with chickpea on chlorophyll *a*, *b*, carotenoid and total chlorophyll of chickpea.**

| | | Chickpea | | | | | | | |
|---|---|---|---|---|---|---|---|---|---|
| | Cropping system | Chl. *a* (mg/g FW) | | Chl. *b* (mg/g FW) | | CAR (mg/g FW) | | Total chl.(mg/g FW) | |
| Irrigation regime | | 2017 | 2018 | 2017 | 2018 | 2017 | 2018 | 2017 | 2018 |
| Normal Irrigation | $c_1$ | 1.25±0.007BCc | 1.11 ±0.005DEbc | 0.33±0.007Bbc | 0.29 ±0.002B-Eabc | 0.23±0.003ABa | 0.22 ±0.004Bab | 1.58±0.005BCb | 1.40±0.008Cb |
| | $c_2$ | 1.27±0.009BCc | 1.09±0.008Ec | 0.27 ±0.006DEc | 0.28 ±0.008CDEbc | 0.16±0.008Dbc | 0.20 ±0.002BCb | 1.54±0.007BCb | 1.37±0.005Db |
| | $b_1c_1$ | 0.79±0.006Fd | 0.72 ±0.005FGd | 0.41±0.003Aa | 0.32 ±0.004BCab | 0.18 ±0.006CDab | 0.17 ±0.006CDbc | 1.20±0.006Ec | 1.04±0.007Fc |
| | $b_1c_2$ | 1.35±0.005Bbc | 1.24±0.004Ca | 0.33±0.002Bb | 0.30 ±0.007BCDab | 0.22±0.008Ba | 0.19 ±0.004CDb | 1.68±0.008Bb | 1.54 ±0.003BCa |
| | $b_2c_1$ | 0.78±0.008Fd | 0.63±0.007Gd | 0.30 ±0.003BCc | 0.25±0.009Ec | 0.09±0.001Ec | 0.11±0.005Ec | 1.08±0.007EFc | 0.88±0.004Gd |
| | $b_2c_2$ | 1.48±0.004Aa | 1.22 ±0.009CDab | 0.39±0.005Aab | 0.33±0.007Ba | 0.23±0.003Aa | 0.26±0.005Aa | 1.87±0.009Aa | 1.55 ±0.007BCa |
| Water stress | $c_1$ | 0.80 ±0.008CDcd | 0.63 ±0.007Ecd | 0.29 ±0.009ABa | 0.21 ±0.006DEbc | 0.22 ±0.002ABCbc | 0.18 ±0.007Cab | 1.09±0.009Cc | 0.84 ±0.002EFcd |
| | $c_2$ | 0.92 ±0.006Bbc | 0.78±0.008Db | 0.30 ±0.007ABa | 0.19 ±0.005EFcd | 0.26±0.004Aa | 0.21 ±0.006ABCa | 1.32±0.004Bb | 0.97 ±0.004CDb |
| | $b_1c_1$ | 0.52±0.004Fe | 0.50 ±0.004Fde | 0.27 ±0.008BCab | 0.23 ±0.004Dabc | 0.11±0.001Dd | 0.09±0.002Dc | 0.79±0.007Fe | 0.73 ±0.008FGde |
| | $b_1c_2$ | 0.72 ±0.007DEd | 0.68 ±0.006Ebc | 0.21 ±0.006DEbc | 0.24 ±0.002CDab | 0.19±0.002BCc | 0.17±0.005Cb | 0.93±0.003DEd | 0.92 ±0.009DEbc |
| | $b_2c_1$ | 0.53±0.004Fe | 0.46±0.003Fe | 0.17 ±0.004FGc | 0.15±0.008Gd | 0.11±0.003Dd | 0.08±0.004Dc | 0.72±0.008FGe | 0.61±0.008Ge |
| | $b_2c_2$ | 1.15±0.009Aa | 1.01±0.008Ba | 0.31±0.007Aa | 0.27±0.004Ba | 0.24 ±0.002ABab | 0.21±0.002Aa | 1.46±0.006Aa | 1.28±0.006Ba |

Chl. a: Chlorophyll *a*; Chl. b: Chlorophyll *b*; CAR: Carotenoid; Total chl.: Total chlorophyll; $c_1$: Sole cropping of chickpea in December; $c_2$: Sole cropping of chickpea in January; $b_1c_1$: Intercropping of barley + chickpea in December; $b_1c_2$: Intercropping of barley in December + chickpea in January; $b_2c_1$: Intercropping of barley in January + chickpea in December; $b_2c_2$: Intercropping of barley + chickpea in January. Means ± SE with common capital letters in each irrigation regime between two years and means ± SE with common lowercase letters in each irrigation regime and column should not be considered different at 0.05 probability based on Tukey-Kramer test.

stress, the late sowing date of chickpea in January intercropped with barley in December ($b_1c_2$) could alleviate the detrimental effects of water stress through increasing CAT and POX activities of chickpea.

## Relative water content (RWC)

In all of the cropping treatments, the RWCs of barley in the 2017 growing season (Fig 2A) were higher than the 2018 (Fig 2B). The higher rainfall, and lower mean temperature and evaporation in 2017 compared to 2018 (Table 2) could explain the seasonal differences in RWC. Also, in each intercropping treatment, RWCs in normal irrigation were more than water stress conditions, significantly ($p \leq 0.05$). In both years of the experiment, the highest RWCs of barley were obtained in $b_1c_2$, $b_1c_1$ and $b_2c_1$ intercropping treatments under normal irrigation regime (Fig 2A and 2B). Similarly, under water stress, barley intercropped with chickpea enhanced the RWCs compared to barley sole cropping. On the other hand, the lowest RWC was observed in $b_2c_2$ with significant differences from the other intercropping treatments. The mean comparison data for RWC of chickpea under different cropping treatments and irrigation regimes during the 2017 and 2018 growing seasons are given in Fig 2C and 2D. In 2017 and 2018, under normal irrigation, the highest RWCs of chickpea were obtained in $b_2c_1$, $b_1c_2$ and $b_1c_1$ which had significant differences with the other treatments. On the other hand, water stress influenced RWC in all of the cropping systems, negatively. In two years, under water stress, intercropping treatments except for the $b_2c_2$ could save more water in their leaves compared to sole cropping of chickpea. It seems that simultaneous sowing of chickpea with barley in January ($b_2c_2$) could not improve the acclimation of chickpea to water stress because of shortening the growing season length and sensitivity of chickpea.

**Table 7. Interaction effect of irrigation regime and cropping system of barley with chickpea on catalase and peroxidase activity of barley.**

| Irrigation regime | Cropping system | CAT (Unit/mg protein) | | POX (Unit/mg protein) | |
|---|---|---|---|---|---|
| | | **2017** | **2018** | **2017** | **2018** |
| Normal irrigation | $b_1$ | 0.40±0.002Fb | 0.45±0.001Fc | 4.7±0.017ABcb | 4.81±0.019ABb |
| | $b_2$ | 0.64±0.003Eb | 0.67±0.004DEc | 1.30±0.009Dc | 1.39±0.008Dd |
| | $b_1c_1$ | 1.48±0.009Ca | 1.56±0.008Cb | 4.11±0.014Cb | 4.21±0.009Cc |
| | $b_1c_2$ | 0.44±0.004Fb | 0.65±0.009Ec | 4.73±0.017ABa | 4.73±0.015ABb |
| | $b_2c_1$ | 2.24±0.007Ba | 2.36±0.008Aa | 4.02±0.015Cb | 4.91±0.018Aab |
| | $b_2c_2$ | 0.51±0.002EFb | 0.61±0.007Ec | 4.33±0.014BCb | 5.12±0.019Aa |
| Water stress | $b_1$ | 0.85±0.007Dc | 0.93±0.009Dd | 5.27±0.011Gd | 6.53±0.018EFd |
| | $b_2$ | 1.48±0.0008Cb | 1.56±0.006BCc | 9.88±0.018Bb | 10.11±0.014Bb |
| | $b_1c_1$ | 1.70±0.009BCb | 1.82±0.008Bbc | 8.26±0.019CDc | 8.91±0.008BCbc |
| | $b_1c_2$ | 1.67±0.007BCb | 1.77±0.007BCbc | 5.41±0.013FGd | 7.23±0.010DEcd |
| | $b_2c_1$ | 2.26±0.011Aa | 2.43±0.011Aa | 9.45±0.017Bbc | 9.56±0.011Bb |
| | $b_2c_2$ | 1.57±0.008BCb | 2.15±0.011Aab | 12.81±0.019Aa | 13.21±0.021Aa |

CAT: Catalase; POX: Peroxidase; $b_1$: Sole cropping of barley in December; $b_2$: Sole cropping of barley in January; $b_1c_1$: Intercropping of barley + chickpea in December; $b_1c_2$: Intercropping of barley in December + chickpea in January; $b_2c_1$: Intercropping of barley in January + chickpea in December; $b_2c_2$: Intercropping of barley + chickpea in January. Means ± SE with common capital letters in each irrigation regime between two years and means ± SE with common lowercase letters in each irrigation regime and column should not be considered different at 0.05 probability based on Tukey-Kramer test.

## Grain yield of barley and chickpea

The interaction effect of the irrigation regime and intercropping treatments of barley with chickpea on grain yield of barley are shown in Fig 3A and 3B. Under normal irrigation, relay intercropping of chickpea with barley ($b_1c_2$), increased the barley grain yield by 18 and 17% compared to sole cropping of barley in December ($b_1$) in 2017 and 2018, respectively. The early establishment of barley in December intercropped with late sowing of chickpea in January improved the competition ability of barley through the faster growth. Under water stress in $b_1c_2$ treatment, grain yield of barley was enhanced 19 and 35% compared to $b_1$ in 2017 and 2018, respectively. In contrast, in another relay intercropping treatment ($b_2c_1$), late sowing of barley could not mitigate adverse effects of water deficit on barley grain yield. Overall, in both of the irrigation regimes, barley grain yield in the first year (Fig 3A) was more than the second year (Fig 3B). In 2017 and 2018, under normal irrigation regime in spite of barley, sole cropping of the chickpea in December and January ($c_1$ and $c_2$) had the highest grain yield in the range of 1.721 to 2.033 t/ha compared to intercropping treatments (Fig 3C and 3D). The grain yield of chickpea in $b_2c_1$ was higher than $c_1$ treatment with no significant difference together, when plants were subjected to water stress.

## Biological yield of barley and chickpea

Different irrigation regimes and cropping systems affected the biological yield of barley and the higher amount of biological yield was obtained in 2017 (Fig 4A) compared to 2018 (Fig 4B). In normal irrigation, the biological yield of $b_1c_2$ intercropping treatment increased 19 and 14% compared to $b_1$ in 2017 and 2018, respectively. A similar trend was observed in water stress conditions, so that biological yield in $b_1c_2$ treatment from 5244 and 4.902 t/ha in $b_1$ enhanced to 6.212 (18% increase) and 5.986 t/ha (22% increase) in 2017 and 2018, respectively (Fig 4A and 4B). Under water stress, the $b_2c_1$ treatment with 3.701 t/ha in 2017 and 3.501 t/ha had no acceptable barley biological yield compared to the other treatments. In normal

**Table 8. Interaction effect of irrigation regime and cropping system of barley with chickpea on catalase and peroxidase activity of chickpea.**

| Irrigation regime | Cropping system | CAT (Unit/mg protein) | | POX (Unit/mg protein) | |
|---|---|---|---|---|---|
| | | 2017 | 2018 | 2017 | 2018 |
| Normal irrigation | $c_1$ | 1.34±0.011Aab | 1.11±0.006Ba | 1.89±0.009Fd | 1.61±0.12Gd |
| | $c_2$ | 0.74±0.009Cc | 0.69±0.005Cb | 3.33±0.015Bb | 2.53±0.011Ec |
| | $b_1c_1$ | 0.45±0.005Da | 0.43±0.006Da | 3.43±0.019Bb | 3.03±0.015CDab |
| | $b_1c_2$ | 1.35±0.009Aab | 1.21±0.008ABa | 2.13±0.017Fc | 2.98±0.009Db |
| | $b_2c_1$ | 0.86±0.007Cc | 0.75±0.005Cb | 4.57±0.024Aa | 3.26±0.012BCab |
| | $b_2c_2$ | 1.21±0.010ABb | 1.11±0.004Ba | 3.32±0.021Bb | 3.43±0.018Ba |
| Water stress | $c_1$ | 1.48±0.008Dc | 1.36±0.007Dc | 5.32±0.025DEcd | 6.28±0.012CDc |
| | $c_2$ | 0.94±0.009Ed | 0.85±0.008Ed | 4.74±0.027Ed | 6.37±0.009CDc |
| | $b_1c_1$ | 1.47±0.0015Dc | 1.23±0.009Dc | 5.68±0.029CDEcd | 6.27±0.013CDc |
| | $b_1c_2$ | 5.22±0.019Aa | 4.31±0.010Ba | 11.43±0.029Aa | 11.9±0.012Aa |
| | $b_2c_1$ | 2.47±0.016Cb | 2.15±0.015Cb | 6.91±0.021BCc | 8.21±0.007Bb |
| | $b_2c_2$ | 2.23±0.018Cb | 2.21±0.018Cb | 9.47±0.017ABb | 10.93±0.015Aa |

CAT: Catalase; POX: Peroxidase; $c_1$: Sole cropping of chickpea in December; $c_2$: Sole cropping of chickpea in January; $b_1c_1$: Intercropping of barley + chickpea in December; $b_1c_2$: Intercropping of barley in December + chickpea in January; $b_2c_1$: Intercropping of barley in January + chickpea in December; $b_2c_2$: Intercropping of barley + chickpea in January. Means ± SE with common capital letters in each irrigation regime between two years and means ± SE with common lowercase letters in each irrigation regime and column should not be considered different at 0.05 probability based on Tukey-Kramer test.

irrigation, sole cropping of chickpea ($c_1$ and $c_2$) increased the biological yield in the range of 4.982 to 5.466 t/ha$^{-1}$ (Fig 4C and 4D). In both years, under water stress, the lowest biological yields of chickpea were obtained in the late sowing of barley and chickpea in January ($b_2c_2$). Overall, the trend of biological yield changes of chickpea in 2018 (Fig 4D) was similar to 2017 (Fig 4C) in both of the irrigation regimes and cropping systems.

## Harvest index of barley and chickpea

In both years, $b_1c_1$ and $b_2$ had the lowest harvest index (HI) of barley compared to the other treatments, under normal irrigation and water stress conditions, respectively (Fig 5A and 5B). Generally, water stress increased the HI of barley and in 2017 was higher than 2018. This is likely due to differences in climatic conditions as previously discussed (Table 2). In normal irrigation, late sowing of chickpea in January ($c_2$), enhanced the HI of chickpea by 39.5% and 39.8% in 2017(Fig 5C) and 2018 (Fig 5D), respectively. In contrast, under water stress, HI in $c_2$ declined sharply as compared to $b_1c_2$ and $b_2c_1$ treatments. The decline of HI in $c_2$ treatment under water deficit might be attributed to shortening the growth period and increasing the intraspecific competition for water in the late season. The HI of chickpea in relay intercropping of barley in January with chickpea in December ($b_2c_1$) enhanced 29.3% in 2017 and 28.3% in 2018 under water stress (Fig 5C and 5D).

## Water use efficiency

The water use efficiency (WUE) in each irrigation regime and cropping system in 2017 and 2018 are presented in Fig 6. Results showed that in all of the treatments WUE in 2017 (Fig 6A) was higher than 2018 (Fig 6B). Interestingly, the highest WUE was obtained in $b_1c_2$ under both of the normal irrigation and water stress in 2017 and 2018 growing seasons. On the other hand, WUE in $c_2$ treatment was less than the other sole cropping treatments. Early sowing of barley in December had a significant effect on WUE as compared to the late sowing date in January ($b_2$). Increasing the WUE in $b_1c_2$ might be related to more acclimation of barley and chickpea to environmental conditions, especially under late season water stress. Also, the higher rainfall and lower mean temperature and evaporation could cause a more WUE in the first year compared to the second year (Table 2).

## Land equivalent ratio

In 2017 and 2018 in each irrigation regime, the LER of barley ($LER_b$) in $b_1c_2$ was higher than the other intercropping treatments in the range of 1.15±0.008 to 1.18±0.008 (Table 9). The LER of chickpea ($LER_c$) in all of the intercropping treatments was higher than 0.5, which demonstrated the advantage of chickpea intercropped with barley as compared to sole cropping of chickpea (Table 9). In both of the irrigation regimes, the lower $LER_c$ of $b_2c_2$ might be attributed to the same sowing date of two crops in January which increases the competition between two crops for water and nutrient uptake. The LER total ($LER_t$) was affected by the irrigation regime and the cropping system, so that in both of the irrigation regimes, in $b_1c_2$ was maximized. In each intercropping treatment, it was no significant difference between the $LER_t$ of 2017 and 2018 growing seasons as well as $LER_b$ and $LER_c$. Also, in all of the treatments, the $LER_b$ was more than $LER_c$, which demonstrated the higher efficiency of barley in absorbing water and light compared to chickpea (Table 9).

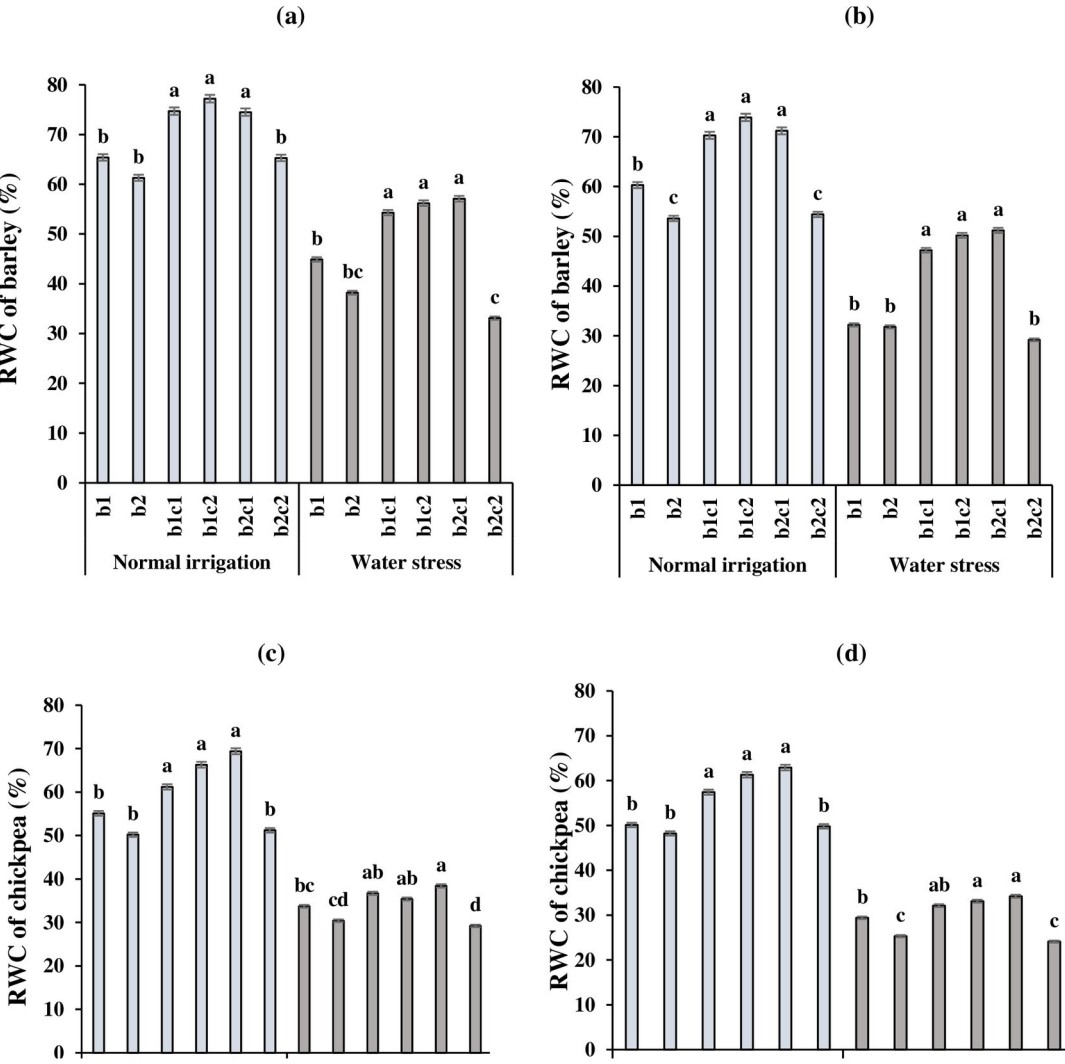

**Fig 2.** Interaction effect of irrigation regime and cropping system of barley with chickpea on relative water content (RWC) of barley in 2017(a) and 2018 (b) and chickpea in 2017(c) and 2018 (d). $b_1$: Sole cropping of barley in December $b_2$: Sole cropping of barley in January, $c_1$: Sole cropping of chickpea in December $c_2$: Sole cropping of chickpea in January, $b_1c_1$: Intercropping of barley + chickpea in December; $b_1c_2$: Intercropping of barley in December + chickpea in January, $b_2c_1$: Intercropping of barley in January + chickpea in December, $b_2c_2$: Intercropping of barley + chickpea in January. In each irrigation regime, values with a letter in common should not be considered different at 0.05 probability based on Tukey-Kramer test. Bars represent mean ± SE.

## Competition ratio

The competition ratio (CR) was influenced by irrigation regime and intercropping treatment, significantly ($p \leq 0.05$) (Table 9). In $b_2c_2$ treatment of two irrigation regimes, CR of barley ($CR_b$) was more than the other intercropping treatment in range of 1.76±0.005 to 1.86±0.006. In $b_2c_1$ under water stress, $CR_b$ declined to 0.80±0.003 in 2017 and 0.71±0.004 in 2018 and differed with $b_1c_2$ and $b_2c_2$ treatments significantly ($p \leq 0.05$). Water stress increased the CR of chickpea ($CR_c$) in $b_1c_1$ and $b_2c_1$, while in $b_2c_2$ declined to 0.56±0.004 in 2017 and 0.54±0.003 in 2018, significantly (Table 9). Late sowing of barley with chickpea simultaneously ($b_2c_2$) in both of the irrigation regimes, enhanced the CR total ($CR_t$) compared to the other treatments

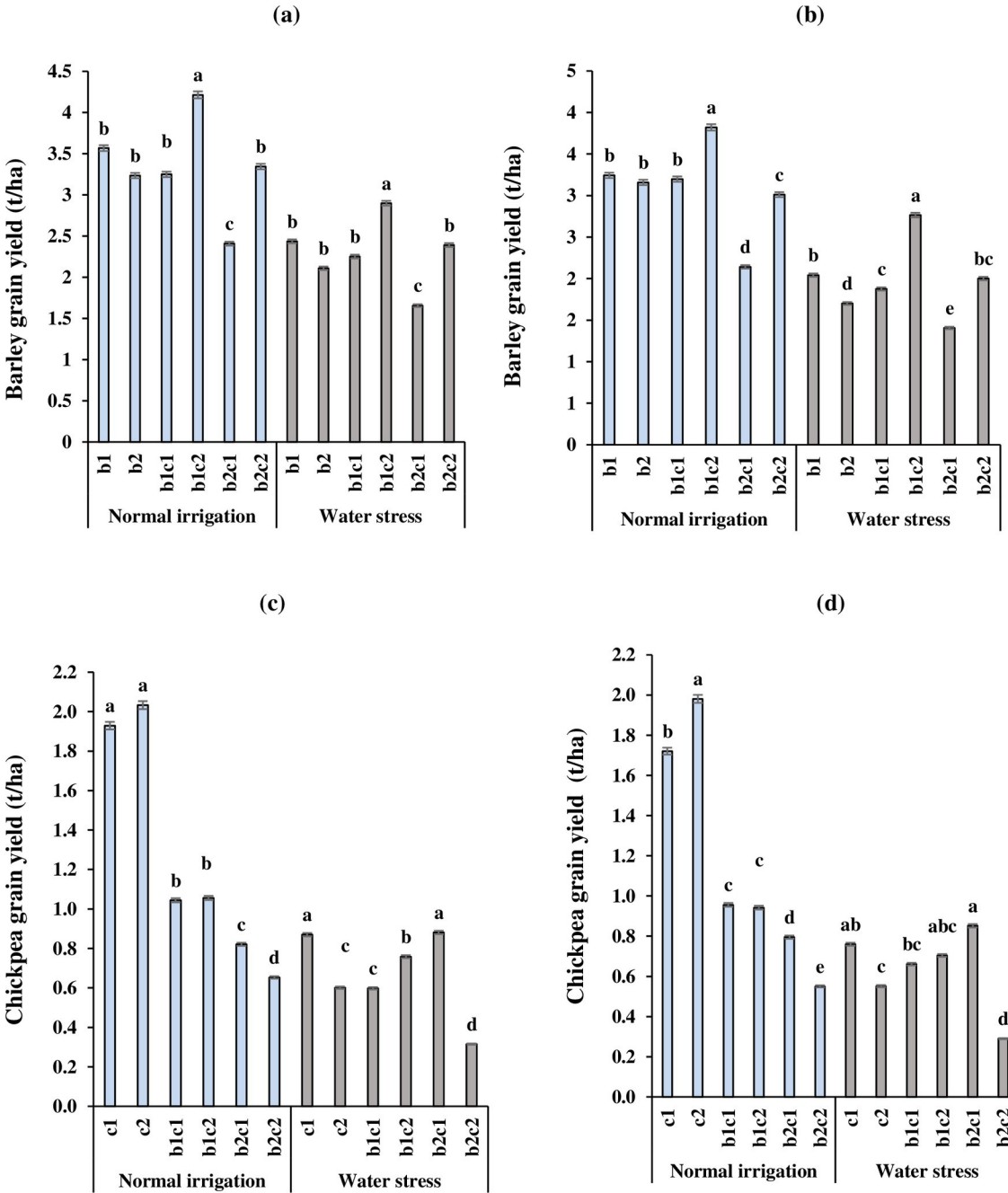

**Fig 3.** Interaction effect of irrigation regime and cropping system of barley with chickpea on grain yield of barley in 2017(a) and 2018 (b) and chickpea in 2017(c) and 2018 (d). $b_1$: Sole cropping of barley in December $b_2$: Sole cropping of barley in January, $c_1$: Sole cropping of chickpea in December $c_2$: Sole cropping of chickpea in January, $b_1c_1$: Intercropping of barley + chickpea in December; $b_1c_2$: Intercropping of barley in December + chickpea in January, $b_2c_1$: Intercropping of barley in January + chickpea in December, $b_2c_2$: Intercropping of barley + chickpea in January. In each irrigation regime, values with a letter in common should not be considered different at 0.05 probability based on Tukey-Kramer test. Bars represent mean ± SE.

(Table 9). In each intercropping treatment, there were no significant differences ($p \leq 0.05$) between the $CR_t$ of 2017 and 2018 growing seasons.

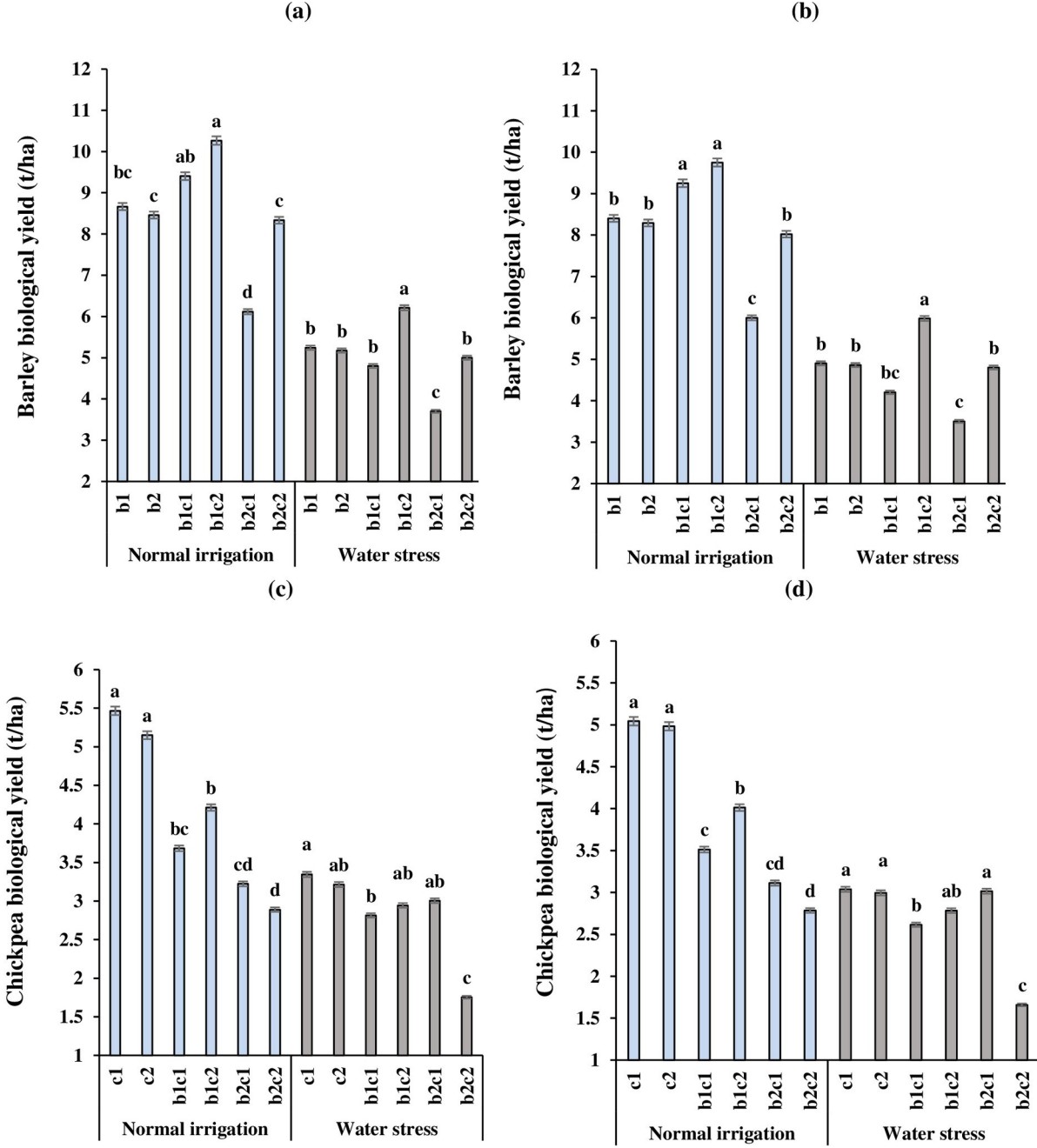

**Fig 4.** Interaction effect of irrigation regime and cropping system of barley with chickpea on biological yield of barley in 2017(a) and 2018 (b) and chickpea in 2017(c) and 2018 (d). $b_1$: Sole cropping of barley in December $b_2$: Sole cropping of barley in January, $c_1$: Sole cropping of chickpea in December $c_2$: Sole cropping of chickpea in January, $b_1c_1$: Intercropping of barley + chickpea in December; $b_1c_2$: Intercropping of barley in December + chickpea in January, $b_2c_1$: Intercropping of barley in January + chickpea in December, $b_2c_2$: Intercropping of barley + chickpea in January. In each irrigation regime, values with a letter in common should not be considered different at 0.05 probability based on Tukey-Kramer test. Bars represent mean ± SE.

## Discussion

In arid areas, water stress is threating agricultural sustainability, and strip-intercropping may serve as a suitable approach to mitigate the challenge. In the south of Iran, with increasing the

temperature in the spring, there was no sufficient rainfall during the reproductive stage of the crop when the water requirement for barley and chickpea enhanced significantly. Because of the occurrence of rainfall in the cool season, the farmers prefer to culture the crops in December. However, they have to irrigate the crop in the warm season [9, 51]. In a similar study on barley- field bean (*Vicia faba* L.) intercropping, Pampana et al. [52] found that the different rainfall amount influenced barley sole crop during two growing seasons. Strip-intercropping increases the spatial distribution of the soil water across the 0–110 cm rooting zones, improves the coordination of soil water sharing during the co-growth period, and provides a compensatory effect for available soil water. The intercropped chickpea used soil water mostly in the top 20-cm layers, whereas cereals were able to absorb water from deeper layers of the neighboring chickpea strips [20]. Chickpea extracts soil water mostly from the shallow (in the top 20 cm) soil depths and the majority of pea roots are concentrated in the 0–30 cm soil profile [5]. In our study, in relay intercropping under water stress, different root distribution in the soil profile, may decrease total water applied ($b_1c_2$ and $b_2c_1$) in comparison to sole cropping of barley ($b_1$ and $b_2$) and chickpea ($c_1$ and $c_2$) (Fig 1).

Chlorophyll content has been applied as a key parameter to evaluate the water status of a crop mainly under water stress [17, 29]. Intercropping enhanced the chlorophyll content of leaves by increasing the nutrient and water availability [22]. Amani Machiani et al. [23] reported that intercropping systems had higher chlorophyll contents compared to sole cropping. Maffei and Mucciarelli [53] declared that peppermint intercropped with soybean created higher chlorophyll and carotenoid contents and biological yield compared to sole cropping. Liu et al. [18] showed that chlorophyll *a* content was decreased sharply in peanut during water stress, while chlorophyll *b* content was approximately constant. Nitrogen, magnesium and zinc, are essential compartments of chlorophyll molecules [54]. Intercropping can enhance the chlorophyll content of leaves by increasing the nutrient availability [22]. Different root systems in cereal-legume intercropping systems is able to improve the mobilization and uptake of nitrogen, and macro and micronutrients effectively, through interspecific interactions in rhizosphere [55, 56]. In the present study, total chlorophyll of barley in all of the cropping treatments decreased significantly ($p \leq 0.05$) when plants were subjected to water stress. In barley, sowing of barley in December intercropped with chickpea in December and January ($b_1c_1$ and $b_1c_2$) created a suitable condition for pigment content enhancement (Table 5). In contrast, in chickpea it seems that simultaneous sowing of barley and chickpea in January ($b_2c_2$) caused a favorite situation in increasing pigment content (Table 6). Singh and Aulakh [5] concluded that the more chlorophyll content in intercropping than sole wheat could be related to nitrogen transfer with chickpea and more soil moisture in wheat intercropped with chickpea.

When crops are exposed to severe water stress, the reactive oxygen species (ROS) like hydrogen peroxide and superoxide accumulated in the leaves. In this condition, crops enhance the antioxidant contents of their leaves to alleviate the negative effects of ROSs [57]. Water stress reduces the photosynthesis and growth rate of crops due to breakdown of the balance between the antioxidant contents such as catalase (CAT), peroxidase enzymes (POX) and ROSs production [58]. Increase in CAT activity is a common response to water stress demonstrate prominent role of CAT in the leaf protection against chlorophyll oxidation [59]. Similar to our results, Mafakheri et al. [51] showed a higher CAT and POX activity under stress in three chickpea genotypes. Nair et al. [60] reported that CAT and POX enhanced significantly in cowpea (*Vigna unguiculata* L.) when crop is exposed to water stress. Little studies have been published in terms of the intercropping effect on antioxidant activity of crops. In one of the few studies, Eskandari and Alizadeh Amraie [9] reported that the interaction effect of the crop system and irrigation regime of Persian clover intercropped with wheat were significant ($p \leq 0.05$) on POX activity and the maximum activity of POX was observed under water stress

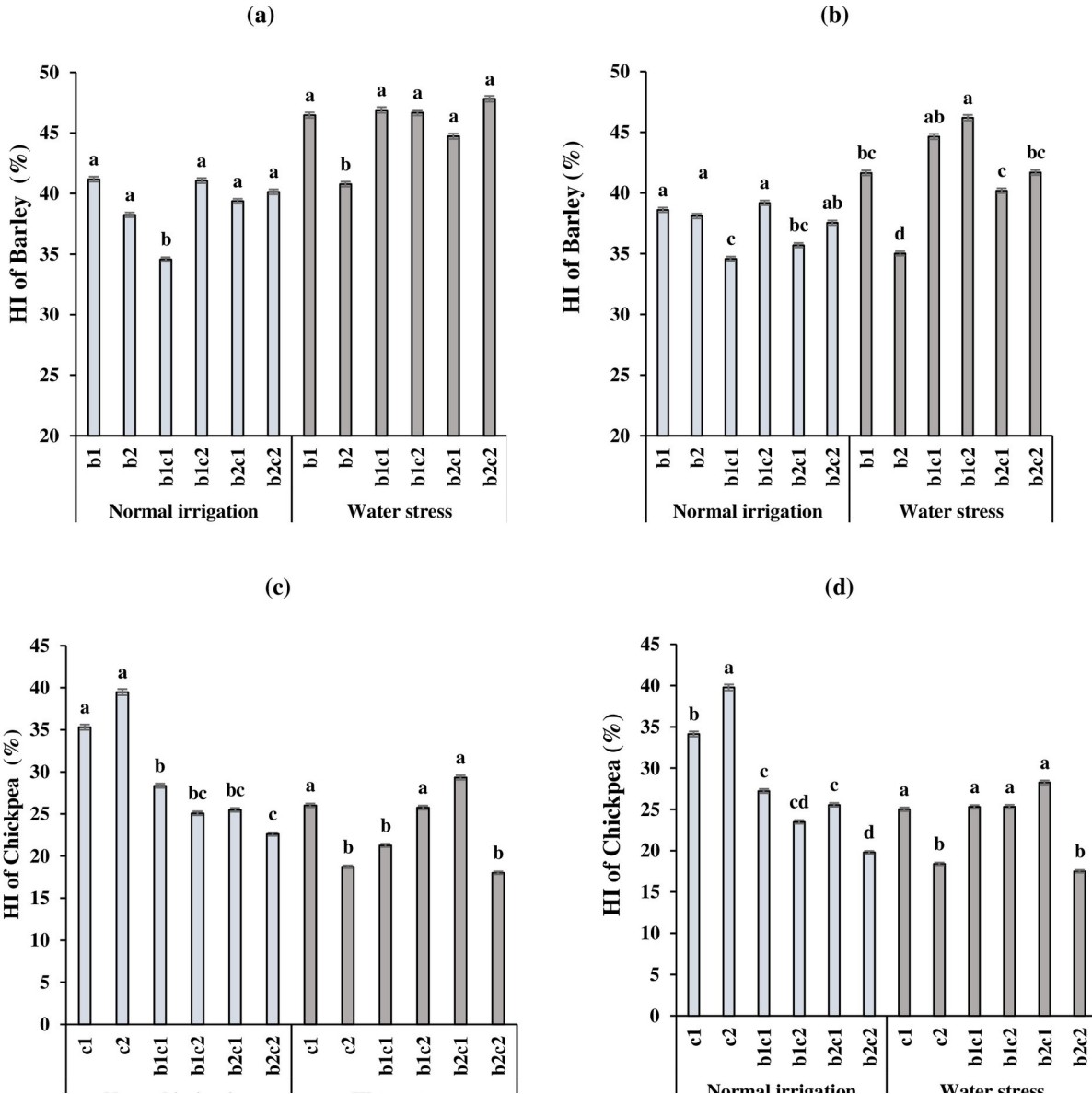

**Fig 5.** Interaction effect of irrigation regime and cropping system of barley with chickpea on harvest index (HI) of barley in 2017(a) and 2018 (b) and chickpea in 2017(c) and 2018 (d). $b_1$: Sole cropping of barley in December $b_2$: Sole cropping of barley in January, $c_1$: Sole cropping of chickpea in December $c_2$: Sole cropping of chickpea in January, $b_1c_1$: Intercropping of barley + chickpea in December; $b_1c_2$: Intercropping of barley in December + chickpea in January, $b_2c_1$: Intercropping of barley in January + chickpea in December, $b_2c_2$: Intercropping of barley + chickpea in January. In each irrigation regime, values with a letter in common should not be considered different at 0.05 probability based on Tukey-Kramer test. Bars represent mean ± SE.

which was 20% higher than wheat. In the present study, relay intercropping of chickpea in January with barley on December ($b_1c_2$) increased the enzyme activity of chickpea more than barley (Tables 7 and 8). This might be attributed to more sensitivity of the chickpea to water deficit and its lower competition ability when intercropped with barley [5, 37].

The RWC represents the water status of a plant, which is related to cell turgidity of the leaves. Also, division and development of cells are closely related to cell turgidity, which influences RWC and grain yield, positively [61, 62]. The tolerant plants maintain more water in

**(a)**

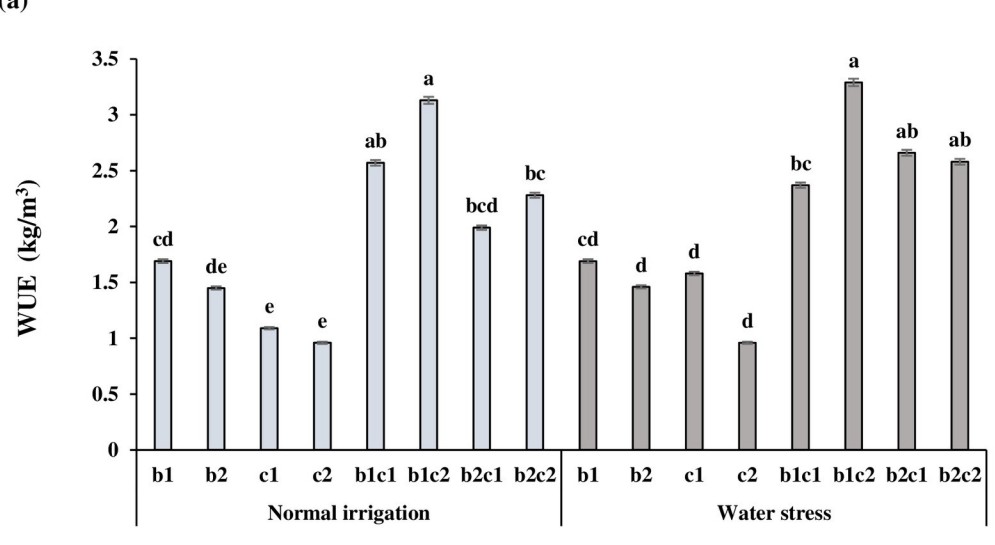

**(b)**

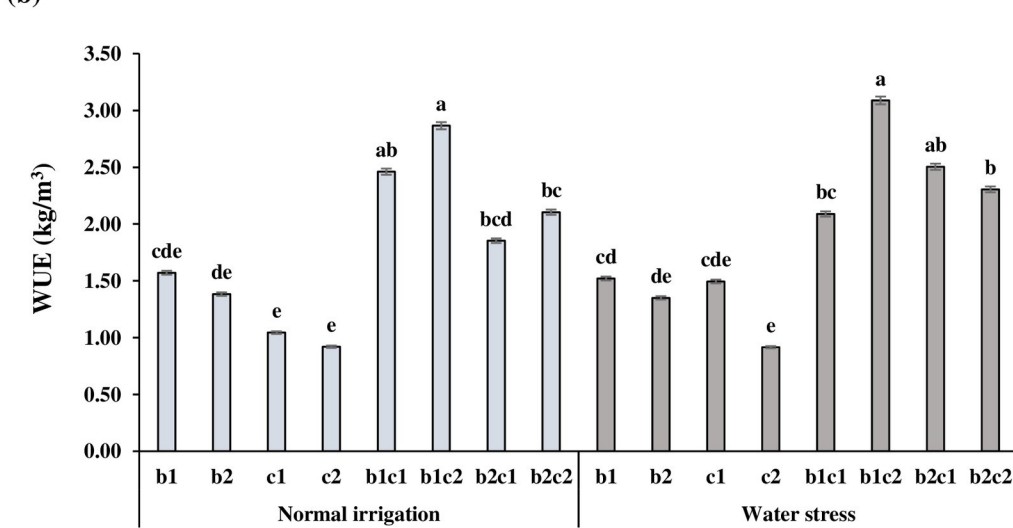

**Fig 6.** Interaction effect of irrigation regime and cropping system of barley with chickpea on water use efficiency (WUE) in 2017 (a) and 2018 (b). $b_1$: Sole cropping of barley in December $b_2$: Sole cropping of barley in January, $c_1$: Sole cropping of chickpea in December $c_2$: Sole cropping of chickpea in January, $b_1c_1$: Intercropping of barley + chickpea in December; $b_1c_2$: Intercropping of barley in December + chickpea in January, $b_2c_1$: Intercropping of barley in January + chickpea in December, $b_2c_2$: Intercropping of barley + chickpea in January. Means sharing the same letter do not differ significantly based on LSD (p≤0.05) test. In each irrigation regime, values with a letter in common should not be considered different at 0.05 probability based on Tukey-Kramer test. Bars represent mean ± SE.

their leaves because of more RWC as compared to sensitive plants [29]. Our findings are in agreement with Eskandari and Alizadeh Amraie [9] who declared that in all cropping systems, the RWC of Persian clover and wheat declined by decreasing available water in the soil under water stress conditions. In our study, RWC in barley is more than chickpea in both of the irrigation regimes and cropping systems, which suggests that barley is more drought tolerant in comparison to chickpea. Also, RWC improved in barley and chickpea when intercropped together. It appears that intercropping barley with chickpea especially in $b_1c_2$ could mitigate

**Table 9. Interaction effect of irrigation regime and cropping system on land equivalent ratio and competition ratio of barley with chickpea intercropping.**

| Irrigation regime | Cropping system | | | | LER | | | | | | CR | | |
|---|---|---|---|---|---|---|---|---|---|---|---|---|---|
| | | Barley | | Chickpea | | Total | | Barley | | Chickpea | | Total | |
| | | 2017 | 2018 | 2017 | 2018 | 2017 | 2018 | 2017 | 2018 | 2017 | 2018 | 2017 | 2018 |
| Normal irrigation | $b_1c_1$ | 1.09 ±0.008ABa | 1.10 ±0.004ABa | 0.67 ±0.003Bb | 0.70 ±0.005ABab | 1.76 ±0.006Cab | 1.80 ±0.009BCab | 1.61 ±0.004ABCab | 1.58 ±0.005BCab | 0.62f ±0.004ABab | 0.63 ±0.007ABab | 2.23c ±0.008ABab | 2.21 ±0.004ABab |
| | $b_1c_2$ | 1.18 ±0.005Aa | 1.16 ±0.003Aa | 0.82 ±0.007Aa | 0.81 ±0.004Aa | 2.00±0.004Aa | 1.97 ±0.008ABa | 1.45±0.007Cbc | 1.44 ±0.006Cab | 0.69 ±0.005ABab | 0.68 ±0.009ABab | 2.14 ±0.004BCab | 2.12 ±0.002BCb |
| | $b_2c_1$ | 0.71 ±0.006Cc | 0.72 ±0.006Cc | 0.59 ±0.006Bb | 0.62 ±0.007Bbc | 1.30 ±0.009Db | 1.34±0.006Dc | 1.20±0.008Dc | 1.17 ±0.004Db | 0.83±0.003Aa | 0.84±0.003Aa | 2.03 ±0.005Cb | 2.01 ±0.006Cb |
| | $b_2c_2$ | 0.98 ±0.003Bb | 0.97 ±0.008Bb | 0.56 ±0.003Bb | 0.54±0.004Bc | 1.55 ±0.007Cab | 1.51 ±0.005CDbc | 1.76±0.005ABa | 1.79 ±0.009Aa | 0.57±0.007Bb | 0.56±0.002Bb | 2.33±0.003Aa | 2.35 ±0.007Aa |
| Water stress | $b_1c_1$ | 0.92 ±0.009BCb | 0.86 ±0.004BCbc | 0.84 ±0.004Aa | 0.86 ±0.008Ab | 1.76 ±0.009Bab | 1.72 ±0.008BCab | 1.09±0.007Cbc | 1.01 ±0.006Cbc | 0.92 ±0.002BCab | 1.02 ±0.004BCab | 2.01±0.002Bb | 2.03 ±0.004Bb |
| | $b_1c_2$ | 1.18 ±0.005Aa | 1.15 ±0.008Aa | 0.92 ±0.003Aa | 0.93 ±0.006Aab | 2.10±0.01Aa | 2.08±0.006Aa | 1.29±0.004Bab | 1.25 ±0.005Bb | 0.77 ±0.005CDbc | 0.80 ±0.008CDbc | 2.07 ±0.004Bab | 2.05 ±0.008Bb |
| | $b_2c_1$ | 0.72 ±0.009Cc | 0.70 ±0.006Cc | 0.90 ±0.007Aa | 0.99 ±0.002Aa | 1.61 ±0.009CDab | 1.69 ±0.007CDb | 0.80±0.003Dc | 0.71 ±0.004Dc | 1.26±0.006Aa | 1.42±0.002Aa | 2.05 ±0.007Bab | 2.12 ±0.006Bab |
| | $b_2c_2$ | 0.97 ±0.003ABb | 0.99 ±0.004ABab | 0.55 ±0.006Bb | 0.53±0.005Bc | 1.51 ±0.008Db | 1.52 ±0.006Db | 1.77±0.006Aa | 1.86 ±0.006Aa | 0.56±0.004Dc | 0.54±0.003Dc | 2.34±0.006Aa | 2.40 ±0.003Aa |

LER: land equivalent ratio; CR: Competition ratio; $b_1c_1$: Intercropping of barley + chickpea in December; $b_1c_2$: Intercropping of barley in December + chickpea in January; $b_2c_1$: Intercropping of barley in January + chickpea in December; $b_2c_2$: Intercropping of barley + chickpea in January. Means ± SE with common capital letters in each irrigation regime between two years and means ± SE with common lowercase letters in each irrigation regime and column should not be considered different at 0.05 probability based on Tukey-Kramer test.

the adverse effects of water stress on crop growth rate by maintaining higher RWC in their leaves. Overall, the higher RWC of barley in $b_1c_1$ and $b_1c_2$ treatments could create a better condition in total chlorophyll increment (Table 5).

In intercropping of cereals with legumes, the yield of each crop might be affected by inter-specific competition for crucial growth resources, allelopathic effects, water stress, sowing date and plant density [1, 5, 63]. Abu-Bakar et al. [12] declared that the highest grain yield of barley was obtained in the sole crop compared to barley intercropped with lentil. In the current study, the lower chickpea grain yield in intercropping could be related to lower competitive ability of chickpea in terms of light, water and nutrients compared to sole cropping (Ahlawat et al. 2005). The higher production in sole cropping can be related to the homogeneous conditions under sole cropping [5]. In contrast, some studies showed that the grain yield and biological yield were improved in intercropping with different crops and environments. For example, De la Fuente et al. [64] declared that sunflower and soybean intercropping created higher grain yield compared to sole cropping, which could be related to complementary use of resources in space and time in intercropping systems. Also, Amossé et al. [65] declared that the wheat canopy had no inhibition effects on seedling establishment of legumes in relay inter-cropping. Känkänen and Eriksson [66] observed that legumes with intercropped barley had the lowest negative effect on barley yield because of slower growth of legumes in the early season. Galanopoulou et al. [34] reported that barley can be grown with faba bean due to crops created a high biological yield and grain yield by exploiting the more resources in comparison to monoculture. Recently, Luhmer et al. [67] reported that barley and poppy (*Papaver somniferum* L.) intercrops produced higher poppy yields compared to sole cropping, whereas early sowing dates of barley enhanced its competition ability. Latati et al. [68] suggested that in a suitable intercropping system, legumes facilitate cereals production through optimum use of environmental resources. Iliadis [69] declared that sown chickpea in autumn or winter created more grain yield than spring sown. In the current study, the early sowing of barley in December intercropped with chickpea in January (b1c2) increased the grain yield (Fig 3A and 3B)

and biological yield (Fig 4A and 4B) of barley due to early suppression of chickpea by vigorous barley. The reduction of chickpea yield in late sowing date of chickpea (December vs. January) might be attributed to a decline in light transmission and interception in the lower levels of canopy, leading to growth and development depression [36]. Also, the lower chickpea grain yield in simultaneous sowing of barley and chickpea in January ($b_2c_2$) might be related to shortening the growth period of chickpea and less its ability to compete with barley, especially under water stress.

Under water stress, greater biological yield of barley in $b_1c_2$ treatments leads to higher grain yield than sole cropping, which is possibly attributed to an increase in the total chlorophyll (Table 5) and (RWC) (Fig 2A and 2B) and light interception by greater canopy at the early sowing date of barley [70]. In contrast, the late sowing date of barley intercropped with chickpea in $b_2c_1$ treatment enhanced chickpea grain yield. Delays in sowing of barley into the intercrop enhanced the suppression ability of chickpea over barley to gain more growth resources. The early sowing date of the legume intercropped with cereals facilitates its seedling establishment, allowing to add more dry matter [65]. The simultaneous sowing date of chickpea with barley in January ($b_2c_2$) decreased the HI of chickpea compared to other intercropping treatments. The late sowing date of chickpea increased the shading of barley on chickpea, which could reduce overall photosynthetic production of chickpea and assimilate allocation to grain decreased, drastically. It appears that in $b_1c_2$ treatment, each crop occupied and accessed to growth resources from different ecological niches at different times due to relay intercropping, while minimizing competitive interactions [10, 65]. Late sowing of chickpea with barley in January ($b_2c_2$), enhanced interspecific competition which reduced the biological yield of chickpea more than barley, especially under water stress. Overall, grain yield, biological yield and HI of chickpea in $b_2c_2$ were less than $b_1c_1$ intercropping treatment because shortening the growing season length due to late season water stress.

The higher LER in intercropping revealed the advantages of intercropping of cereals and legumes because of better utilization of resources like light, nutrient uptake and water [32, 34, 71]. Galanopoulou et al. [34] declared that in all intercrop treatments, the LERs of barley were higher than 0.5, while in faba bean were lower than 0.5, which demonstrated the advantages of barley compared to the faba bean. Hauggaard-Nielsen et al. [72] reported that in pea intercropped with barley, the LER of chickpea declined, while the partial LER of barley increased significantly. In our study, LER amounts of barley and chickpea were more than 0.5 while in all of the intercropping treatments, the LER of barley was higher than chickpea. Similar to our results, Hamzei and Seyedi [73] declared that in all intercropping treatments of barley with chickpea, the LER total was higher than one. It's demonstrated the superiority of relay intercropping especially in $b_1c_2$ treatments compared to sole cropping. Eskandari and Alizadeh Amraie [9] in a similar study declared that the LER of wheat-Persian clover intercropping under water stress was more than normal irrigation. They suggest that intercropping mitigates the detrimental effects of the water deficit by RWC and WUE enhancement of each crop, and finally, it leads to an increase in LER. Chen et al. [20] in a three-year experiment reported that corn-chickpea intercropping increased corn grain yield by an average of 25% and enhanced WUE of corn by 24%. In dry areas with high soil evaporation, increasing biomass production and WUE of intercropping system is partly attributable to water sharing through possible water movement between the rooting zones and water compensation from one strip to the other. Fan et al. [74] reported that corn–pea intercropping produced 23–38% greater total yield than corresponding sole crops, as the LER ranged from 1.23 to 1.38 and intercropping improved water capture by plants compared to sole corn. In the current study, one of the reasons for increasing LER$_t$, especially under water stress, in intercropping of barley in December

+ chickpea in January ($b_1c_2$) was enhancing the grain yield (Fig 3A), RWC (Fig 2A) and WUE (Fig 6) of barley compared to sole cropping.

The CR reveals a useful evaluation of competition ability between two crops in intercropping [75]. Amani Machiani et al. [23] found that the CR amount of peppermint was more than 1, which was higher than soybean, showing a yield superiority of peppermint intercropped with soybean. Andrade et al. [75] suggested that avoiding the overlapping of critical growth stages by relay intercropping improves resources use between intercrop components. They also found that the yield advantage in intercrop is lower under scarce water availability and mainly associated with a decrease in intercropped legume productivity. Veisi et al. [14] declared that the chickpea had weak competition ability compared to the other plants because a slow growth rate, especially in the seedling establishment stage. The more competition ability of cereals intercropped with legumes may be related to that the fact that cereals take up more water and nitrogen in the early season and accumulate more dry matter, which cause shad on the legume and thereby reduce its competition ability in the intercropping system [34, 72]. The more CR of barley compared to chickpea in all intercropping systems and irrigation regimes except for the $b_2c_1$ treatments, was consistent with the results of Megawer et al. [76]. In our study, the higher CR of barley compared to chickpea demonstrated the higher aggressivity and superiority of barley and its capability in taking up more resources compared with chickpea.

## Conclusions

Relay intercropping affected the photosynthetic pigment, antioxidant activities and yield of barley and chickpea. As a result, relay intercropping of barley-chickpea tended to be more productive as early sowing of barley in December is intercropped with late sowing of chickpea in January. In relay intercropping, using crops with different rooting structures and suitable sowing dates, productivity can be improved through enhancing biochemical properties, relative water content and water use efficiency. It is concluded that relay intercropping of barley in December with chickpea in January can be a suitable intercropping system for sustainable agriculture, under water stress. Further researches is recommended to investigate the effect of different ratio of barley-chickpea intercropping on biochemical properties and yield under water stress.

## Supporting information

**S1 File.**
(XLSX)

## Acknowledgments

The authors thank to Abdullah Setodeh for his assistance in the lab, as well as the Agriculture and Natural Resources Research Center of Darab, Fars Province, Iran for providing the seeds of this research.

## Author Contributions

**Conceptualization:** Ehsan Bijanzadeh.

**Data curation:** Negin Mohavieh Assadi.

**Formal analysis:** Negin Mohavieh Assadi.

**Funding acquisition:** Ehsan Bijanzadeh.

**Investigation:** Negin Mohavieh Assadi.

**Methodology:** Negin Mohavieh Assadi.

**Project administration:** Negin Mohavieh Assadi, Ehsan Bijanzadeh.

**Resources:** Negin Mohavieh Assadi.

**Software:** Negin Mohavieh Assadi, Ehsan Bijanzadeh.

**Supervision:** Ehsan Bijanzadeh.

**Validation:** Negin Mohavieh Assadi.

**Visualization:** Negin Mohavieh Assadi, Ehsan Bijanzadeh.

**Writing – original draft:** Negin Mohavieh Assadi, Ehsan Bijanzadeh.

**Writing – review & editing:** Ehsan Bijanzadeh.

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
