## [Decision Letter · Decision Letter 0]

8 Nov 2022

PONE-D-22-21858Influence of relay intercropping of barley with chickpea on biochemical characteristics and yield under water stressPLOS ONE

Dear Dr. Bijanzadeh,

Thank you for submitting your manuscript to PLOS ONE. After careful consideration, we feel that it has merit but does not fully meet PLOS ONE’s publication criteria as it currently stands. Therefore, we invite you to submit a revised version of the manuscript that addresses the points raised during the review process.

 See the additional editor comments section for further explanation.

We look forward to receiving your revised manuscript.

Kind regards,

Sergio Saia, Ph.D.

Academic Editor

PLOS ONE

Journal Requirements:

Additional Editor Comments:

Dear authors,

I agree with both reviewers that the ms is interesting but badly handled by the statistical point of view. In particular, the statistical analysis is not replicable and application of a LSD post hoc to such a complex design will bring you to high chances of false positive results. 

I thus suggest you to carefully take into account the revierwers' comment and make one again a protected statistical analysis. Since you can use SAS, I can also provide you a suggestion, as follows:

**BEFORE RUNNING IT, DELETE THE EXPLANATION WITHIN 2 ASTERISKS, ALSO, PLEASE MIND THAT i CALLED THE IRRIGATION AS 'IR' AND THE CROPPING SYSTEM AS 'CS', REPL IS FOR REPLICATES**

proc glimmix data=work.NAME_OF_FILE;

title ‘NAME OF YOUR EXPERIMENT’; ***you can delete this line if you want***

class IR CS Repl Year;

model YOUR_VARIABLE = IR|CS/ DDFM=KR; ***The the symbol | is not an i, and will also produce the interactions terms, you can also split it in 3 treatments, e.g. taking into account the time of intercropping, but it will make your analysis definitely more complex and hard to discuss. KR will correct the estimate of the variances and of the standard errors***

random Year Rep(Year);
***year must be a random statement and replication within year too. Replication per se is not a random statement since the experiment is not replicated in the same plots of the previous year. This will make your analysis simpler, providing a rebuilt least squared means across years. In figs and tabs, you will thus report the year and replicate means, not the results by year, which is not correct nd was used at the time of McIntosh 1983, but not now after Littel et al 1992 with mixed models***

LSMEANS  IR|CS / adjdfe=row adjust=tukey pdiff lines; ***this will adjust for multiple comparison and provide you the classical "letter statements. Please note that LSmeans may differ from your means depending on the probability density distribution of your data and presence of unbalancing in the dataset. LSmeans are "**predicted population margins; that is, they estimate the marginal means over a balanced population.**" [from the SAS guidelines. Adjust=Tukey does not adjust multiple comparisons for the post-hoc tukey, but for the post-hoc of tukey-kramer. This is more advisable when the total comparisons are more than 6, as in your experiment"***

run;

quit;

In addition to the suggestion above, I suggest you to change the sentence "Means sharing the same letter do not differ significantly based on LSD (p≤0.05) test." to "Means with a letter in common should not be considered different according to the post-hoc test at p≤0.05".

Lastly, I mind you that Plos requires for the row data to be published. You should upload at least the means and standard deviations or errors (to be specified along the number of replicates) in the supplementary materials or other repository and indicate it in the ms.

Regards

Sergio.

Reviewers' comments:

Reviewer's Responses to Questions

**Comments to the Author**

1. Is the manuscript technically sound, and do the data support the conclusions?

Reviewer #1: Partly

Reviewer #2: No

2. Has the statistical analysis been performed appropriately and rigorously? 

Reviewer #1: I Don't Know

Reviewer #2: No

3. Have the authors made all data underlying the findings in their manuscript fully available?

Reviewer #1: Yes

Reviewer #2: Yes

4. Is the manuscript presented in an intelligible fashion and written in standard English?

Reviewer #1: No

Reviewer #2: No

5. Review Comments to the Author

Reviewer #1: While the research carried out might bring some valuable insights into the intercropping (and relay cropping) performance under water stress conditions, the manuscript presents several weaknesses that must be addressed. I provide some general comments for each part of the manuscript in this section:

General: the English level of the manuscript is not adequate for a scientific manuscript. There are several grammar mistakes, misspellings, word choices and sentence constructions that make the text unintelligible at some points.

Introduction: the ideas expressed are not well connected.

I lacked some references and explanations relating to the relationship between pigment contents and crop performance and how intercropping affects these variables.

M&M: please, consider the size of plots in future research, 3x2 m is quite a small size for intercrop trials. Plots of this size tend to overestimate the yield that a farmer would get in a commercial field.

You do not state the application field of your research: horticultural, field crops, etc. If field cropping is the target of your research, the use of sowing machines should be considered rather than hand sowing, as the latter does not represent the conditions a farmer might face in a field.

The statistical analyses are not described in sufficient detail to be replicated by other researchers or to correctly interpret the results.

Results: The result display is quite poor. The significative triple interaction in all the variables led to several Figures and Tables that are quite difficult to comprehend from an external reviewer. In addition, the result section does not provide clear and key results of this research.

Discussion: the English level in which the discussion is written does not allow an objective evaluation of it, as some mistakes (grammar or word choice) can lead to incorrect judgement.

Conclusions: some results are summarized in this section, but no clear conclusions are stated.

Reviewer #2: The manuscript “Influence of relay intercropping of barley with chickpea on biochemical characteristics and yield under water stress” aimed to investigate the effects of water stress on barley-chickpea relay intercropping. The evaluation focused both on agronomic performance and biochemical parameters of crops.

On the whole, the topic of this paper is very interesting. Despite the effects of the co-cultivation between cereals and grain legumes on biochemical aspects of crops have been rarely investigated in previous study on intercropping, they are fundamental for better understand the interactions between crops in a more diversified cropping system. Therefore, I appreciate the novelty of this manuscript.

The introduction provides sufficient background, even if the number of relevant references can be increased. The research is design appropriately and methods are adequately described (except for the statistics section). I’m convinced that authors of this manuscript have extremely interesting data however they failed to valorize them in this manuscript. In fact, the results are not clearly presented, a lot of details are missing, and the manuscript is very difficult to read. Moreover, results can be misinterpreted due to the inappropriate statistical analysis.

General comments:

English language and style: The manuscript is not easy to read, an extensive editing of English language and style is required.

Statistical analysis: statistical analysis needs to be revised. I have several concerns related with the data presented in table 5, 6, 7, 8 and figures. Standard errors are missing everywhere, and unit of measures are often wrong. Please see specific comments below.

Data presentation: Tables and figure needs to be deeply revised according to the revised statistical analysis.

Specific comments:

Line 47: Please add a reference “…another crop (cit).

Line 53: Please add a reference “…of the world (cit).

Line 53: Please add a reference “…final yield (cit).

Line 84: “At the best of our knowledge” instead of “Until now”

Line 98: Please add reference defining the soil type e.g “Jahn R, Blume H, Asio V, Spaargaren O, Schad P (2006) Guidelines for soil description, 4th edn. FAO, Rome”

Line 126: Please, define the plowing depth.

Line 129: Define inter-rows space.

Line 144: for barley and chickpea the effective root zone is around 20 -25 cm, why did you consider 90 cm?

Table 3 and 4: In order to improve the tables readability, please define the meaning of I and C in the caption.

Lin 147: Is it a common practice irrigating barley in Iran? Or it is just ax experimental practice?

Line 311: “mg. mg. g-1 “ is not the correct annotation for the unit of measure. The correct annotation is “mg · g-1”. Please check all the unit of measure in the text and SE should be also included.

Everywhere in the text: please, you should include standard error (SE) when you report data in the text.

TABLES

Table 1:

• “Soil organic matter (SOM)” instead of “Organic content”

• I would express N as g/kg

• P, is it total P or Olsen P? Same for the nitrogen, which kind of protocol did you follow for the Nitrogen determination? Kjeldahl method? Please add this information in table caption.

Table 5: You mentioned that “year” and “irrigation regime” are significant factors. In this case you should perform the statistical analysis separately for each year and for each irrigation regime. In the table you have assigned letters of significance in the wrong way. Please perform a post hoc separately for year and irrigation regime and eventually assign capital letter for indicating differences between years. You should include standard error (SE).

Table 6: see comments above

Table 7: see comments above

Table 8: see comments above

Line 362: “Unit. mg-1 protein” “Unit· mg-1 protein”

FIGURE

Figure 1: You should perform post-hoc of your model separately for 2017 and 2018. Use lowercase letter for 2017 and uppercase letter for 2018. SE are missing, please provide them. In case that year is not a significant factor you could present data as average between 2017 and 2018 for each of the treatments. Another option can be present data on 2017 and 2018 in two separate graphs.

Same for figure 2, 3, 4, 5, 6, 7.

6. PLOS authors have the option to publish the peer review history of their article (what does this mean?). If published, this will include your full peer review and any attached files.

Reviewer #1: No

Reviewer #2: No

---

## [Author Response · Author response to Decision Letter 0]

25 Dec 2022

Dear Editor of Plos One

I sent the tracked-change revision of our manuscript entitled ''Influence of relay intercropping of barley with chickpea on biochemical characteristics and yield under water stress'' (PONE-D-22-21858) as well as Response to Reviewers by attachments. Also, I replied to reviewer's comments as below:

Additional Editor Comments:

Dear authors,I agree with both reviewers that the ms is interesting but badly handled by the statistical point of view. In particular, the statistical analysis is not replicable and application of a LSD post hoc to such a complex design will bring you to high chances of false positive results.

Author response: I used Tukey test instead of LSD throughout the manuscript.

In addition to the suggestion above, I suggest you to change the sentence "Means sharing the same letter do not differ significantly based on LSD (p≤0.05) test." to "Means with a letter in common should not be considered different according to the post-hoc test at p≤0.05". 

Author response: See all of the tables and figures., please.

Lastly, I mind you that Plos requires for the row data to be published. You should upload at least the means and standard deviations or errors (to be specified along the number of replicates) in the supplementary materials or other repository and indicate it in the ms. 

Author response: Means data ±SE are given throughout the manuscript. 

Comments to the Author

1. Is the manuscript technically sound, and do the data support the conclusions?

Reviewer #1: Partly

Reviewer #2: No

Author response: See the text, please. The conclusion section has been revised, as well. 

2. Has the statistical analysis been performed appropriately and rigorously?

Reviewer #1: I Don't Know

Reviewer #2: No

Author response: The statistical analysis changed and cropping system in each irrigation regime and year compared, separately. Also, the means were compared by Tukey's test.

3. Have the authors made all data underlying the findings in their manuscript fully available?

Reviewer #1: Yes

Reviewer #2: Yes

4. Is the manuscript presented in an intelligible fashion and written in standard English?

Reviewer #1: No

Reviewer #2: No

Author response: The manuscript has been revised. See the text, please.

Reply to Reviewer #1: While the research carried out might bring some valuable insights into the intercropping (and relay cropping) performance under water stress conditions, the manuscript presents several weaknesses that must be addressed. I provide some general comments for each part of the manuscript in this section:

• General: the English level of the manuscript is not adequate for a scientific manuscript. There are several grammar mistakes, misspellings, word choices and sentence constructions that make the text unintelligible at some points.

Author response: The English level of the manuscript has been improved.

• Introduction: the ideas expressed are not well connected. I lacked some references and explanations relating to the relationship between pigment contents and crop performance and how intercropping affects these variables. Author response: Some references were added to this section. See the Introduction section, please.

• M&M: please, consider the size of plots in future research, 3x2 m is quite a small size for intercrop trials. Plots of this size tend to overestimate the yield that a farmer would get in a commercial field. 

Author response: Yes. You are right.

You do not state the application field of your research: horticultural, field crops, etc. If field cropping is the target of your research, the use of sowing machines should be considered rather than hand sowing, as the latter does not represent the conditions a farmer might face in a field.

Author response: The relay intercropping system is more used for smallholder farmers and traditional agriculture in dry and tropical areas for most plants. Therefore, in this system, most of the plants are cultivated manually [2, 32].

The statistical analyses are not described in sufficient detail to be replicated by other researchers or to correctly interpret the results.

Author response: The statistical analysis changed and each irrigation regime and year compared, separately. Also, the means were compared by Tukey test.

Results: The result display is quite poor. The significative triple interaction in all the variables led to several Figures and Tables that are quite difficult to comprehend from an external reviewer. In addition, the result section does not provide clear and key results of this research.

Author response: Data was analyzed by SAS software 2012 (version 9.4) and the means were compared Tukey test at 0.05 probability level (p≤ 0.05). Because of significant effect of year × irrigation regime × cropping system on considered traits, the data of two years for barley and chickpea presented, separately. This section revised, again. 

Discussion: the English level in which the discussion is written does not allow an objective evaluation of it, as some mistakes (grammar or word choice) can lead to incorrect judgement.

Author response: See the Discussion section, please.

Conclusions: some results are summarized in this section, but no clear conclusions are stated.

Author response: The Conclusions section has been revised.

Reply to Reviewer #2: The manuscript “Influence of relay intercropping of barley with chickpea on biochemical characteristics and yield under water stress” aimed to investigate the effects of water stress on barley-chickpea relay intercropping. The evaluation focused both on agronomic performance and biochemical parameters of crops.

On the whole, the topic of this paper is very interesting. Despite the effects of the co-cultivation between cereals and grain legumes on biochemical aspects of crops have been rarely investigated in previous study on intercropping, they are fundamental for better understand the interactions between crops in a more diversified cropping system. Therefore, I appreciate the novelty of this manuscript.

The introduction provides sufficient background, even if the number of relevant references can be increased. The research is design appropriately and methods are adequately described (except for the statistics section). I’m convinced that authors of this manuscript have extremely interesting data however they failed to valorize them in this manuscript. In fact, the results are not clearly presented, a lot of details are missing, and the manuscript is very difficult to read. Moreover, results can be misinterpreted due to the inappropriate statistical analysis.

Author response: In the introduction section, some new refences were added. Also, data was analyzed by SAS software 2012 (version 9.4) and the means were compared Tukey test at 0.05 probability level (p≤ 0.05). Because of significant effect of year × irrigation regime × cropping system on considered traits, the data of two years for barley and chickpea presented, separately. 

• General comments:

English language and style: The manuscript is not easy to read, an extensive editing of English language and style is required.

Author response: The English level of the manuscript has been improved.

• Statistical analysis: statistical analysis needs to be revised. I have several concerns related with the data presented in table 5, 6, 7, 8 and figures. Standard errors are missing everywhere, and unit of measures are often wrong. Please see specific comments below. 

Data presentation: Tables and figure needs to be deeply revised according to the revised statistical analysis.

Author response: The Standard errors were added in all of the tables and figures. Also, unit of measures has been checked, again.

Specific comments:

Line 47: Please add a reference “…another crop (cit).

Author response: See the text, please.

Line 53: Please add a reference “…of the world (cit). 

Author response: See the text, please.

Line 53: Please add a reference “…final yield (cit). 

Author response: I didn't find this sentence in line 53.

Line 84: “At the best of our knowledge” instead of “Until now”.

Author response: See the text, please.

Line 98: Please add reference defining the soil type e.g “Jahn R, Blume H, Asio V, Spaargaren O, Schad P (2006) Guidelines for soil description, 4th edn. FAO, Rome”. 

Author response: See the text, please.

Line 126: Please, define the plowing depth. 

Author response: The plowing depth was 30 cm.

Line 129: Define inter-rows space.

Author response: The inter-row space between barley and chickpea was 30 cm.

Line 144: for barley and chickpea the effective root zone is around 20 -25 cm, why did you consider 90 cm? 

Author response: The effective root zone is the depth within which most crop roots are concentrated, which was estimated as ∼50–100 cm for barley and as ∼60–70 cm for chickpea (Fan et al., 2016). Thus, with respect to monitor the soil water content at root zone the soil water content was traced in each plot at 30 cm depth down to 90 cm, gravimetrically. 

Table 3 and 4: In order to improve the tables readability, please define the meaning of I and C in the caption. 

Author response: Irrigation regimes (I) included of normal irrigation and cutting off irrigation at full anthesis stage of barley and cropping system (c) consisted of sole and delay intercropping of barley and chickpea in December and January. See the text, please.

Lin 147: Is it a common practice irrigating barley in Iran? Or it is just an experimental practice? 

Author response: Yes. it is a common practice in barley. Unfortunately, in some years, there is no considerable rainfall in March to May when the water requirement of safflower increased to complete the seed filling period (Table 2). These conditions usually are typical in sought of Iran which has dry and hot spring season. With respect to occurrence of rain fall in the cool season, the farmers have to irrigate the crop after anthesis.

Line 311: “mg. mg. g-1 “is not the correct annotation for the unit of measure. The correct annotation is “mg · g-1”. Please check all the unit of measure in the text and SE should be also included. 

Author response: All of the units were checked throughout the manuscript.

Everywhere in the text: please, you should include standard error (SE) when you report data in the text. 

Author response: See the text, please. 

TABLES

Table 1:

• “Soil organic matter (SOM)” instead of “Organic content”

Author response: See the text, please.

• I would express N as g/kg. 

Author response: That's converted to g/kg in the Table 1.

• P, is it total P or Olsen P? Same for the nitrogen, which kind of protocol did you follow for the Nitrogen determination? Kjeldahl method? Please add this information in table caption.

Author response: P content of the soil is Olsen P and N content of the soil was determined by the Kjeldhal method. 

Table 5: You mentioned that “year” and “irrigation regime” are significant factors. In this case you should perform the statistical analysis separately for each year and for each irrigation regime. In the table you have assigned letters of significance in the wrong way. Please perform a post hoc separately for year and irrigation regime and eventually assign capital letter for indicating differences between years. You should include standard error (SE).

Table 6: see comments above

Table 7: see comments above

Table 8: see comments above

Author response: In all of the tables, each irrigation regime and year were analyzed, separately. Means with common capital letters in each irrigation regime within two years and means with common lowercase letters in each irrigation regime and year indicate no significant difference based on Tukey's test at p≤0.05. Vertical bars represent ±SE. 

Line 362: “Unit. mg-1 protein” “Unit· mg-1 protein”.

Author response: That's converted to Unit. mg-1 protein throughout the manuscript

FIGURE

Figure 1: You should perform post-hoc of your model separately for 2017 and 2018. Use lowercase letter for 2017 and uppercase letter for 2018. SE are missing, please provide them. In case that year is not a significant factor you could present data as average between 2017 and 2018 for each of the treatments. Another option can be present data on 2017 and 2018 in two separate graphs.

Same for figure 2, 3, 4, 5, 6, 7.

Author response: In all of the figures, data for each year was presented, separately.

Sincerely,

Corresponding author

---

## [Decision Letter · Decision Letter 1]

20 Mar 2023

PONE-D-22-21858R1Influence of relay intercropping of barley with chickpea on biochemical characteristics and yield under water stressPLOS ONE

Dear Dr. Bijanzadeh,

Thank you for submitting your manuscript to PLOS ONE. After careful consideration, we feel that it has merit but does not fully meet PLOS ONE’s publication criteria as it currently stands. Therefore, we invite you to submit a revised version of the manuscript that addresses the points raised during the review process. in particular, see my opinion in the "additional editor's comments".

We look forward to receiving your revised manuscript.

Kind regards,

Sergio Saia, Ph.D.

Academic Editor

PLOS ONE

**Additional Editor Comments:**

Deal authors, one of the 2 former reviewers accepted to re-review your ms. I frankly have to say that I almost agree with the reviewer. In particular, the english language can be improved by yourself trying to use simpler sentences and making a general polishing of the work. So far, my greatest worry is about the statistical analysis and the reviewer pointed to the need of ensuring its reliability, by suggesting the GLMM (which, by the way, can also handle non normal distributed data o residues and few replications). I please exhort you to carefully take into account that the statistical analysis should be done reducing the false positive and false negative results, with a certain degree of protection of the results given by proper statistics (e.g. Benjamini and Hochberg or Tukey-Kramer, etc.)

Regarding the statistics, please pass over to the indication of the rev. regarding "Values with different letters are not significantly different at 0.05 confidence level." The correct statement is "Values with a letter in common should not be considered different at 0.05 probability.", which poses an additional degree of protection of the results.

Reviewers' comments:

Reviewer's Responses to Questions

**Comments to the Author**

1. If the authors have adequately addressed your comments raised in a previous round of review and you feel that this manuscript is now acceptable for publication, you may indicate that here to bypass the “Comments to the Author” section, enter your conflict of interest statement in the “Confidential to Editor” section, and submit your "Accept" recommendation.

Reviewer #2: (No Response)

2. Is the manuscript technically sound, and do the data support the conclusions?

Reviewer #2: Partly

3. Has the statistical analysis been performed appropriately and rigorously? 

Reviewer #2: No

4. Have the authors made all data underlying the findings in their manuscript fully available?

Reviewer #2: Yes

5. Is the manuscript presented in an intelligible fashion and written in standard English?

Reviewer #2: No

6. Review Comments to the Author

Reviewer #2: The authors mentioned that the English has been revised in the manuscript but it is seems not: the paper is difficult to read, especially for the results section. Moreover, there are too many tables and figures. I would suggest removing some tables or move them in supplementary material and to add relevant information in the main text. Figures 3, 4 and 5 for instance can be merged in 1 graph. I still have some concerns about statistical analysis (see specific comments below). I’m convinced that your data and research topic is interesting however, I would suggest the authors for an extensive English check otherwise the MS could be not considered for the publication.

See specific comments below.

Abstract:

Line 25: “as split plot based on a randomized complete block design” I would remove this information from the abstract.

Line 37-30: I would suggest removing this detailed sentence from the abstract, it is more appropriate for materials and methods section. It is enough informative if you mention that the second factor was the sowing time (December vs January).

Line 31: “Chlorophyll a content of barley increased in b1c2, by consuming less water compared to sole cropping.” Better if you explain it in terms of…In case of relay intercropping, the early establishment of barley determined an higher chlorophyll content in biomass compared to… due to the… Please try to be more precise in your statement, why b1c1 consumed less water? Less competition from chickpea? Early establishment ensured a growth advantage and therefore a better use of water?

Line 33: What do you mean with “(b1c1 and b1c2) created a suitable canopy to pigment contents improvement”.

Line 34 “carotenoid content of chickpea” please specify if you are referring to chickpea biomass or grains.

Line 35-36 “Barley-chickpea intercropping reacted to water deficit through enzymes activity, water use efficiency and land equivalent enhancement compared to sole cropping”. The WUE and LER are not “reactions”. It is better saying that intercropping enhanced the WUE and guaranteed a more efficient use of space (LER>1) compared with sole crops.

Line 36-38 “Under water stress, in b1c2, by increasing total chlorophyll and water use efficiency, grain yield of barley enhanced compared to b1”. This sentence is not clear at all.

Line 38: “It seems”. Not appropriate for abstract

Introduction:

My impression is that sentences are not linked to each other. The introduction needs to be rephased in order to improve the readability of the paper. Moreover, make clear your objectives and hypothesis. I strongly suggest submitting the manuscript for an extensive English revision by experts.

Line 104: carried out instead of “laid out”

Line 127: Is it Darab cv a registered cultivar of chickpea or an ecotype?

Line 132: “which is suitable for hot and dry areas” better say suitable for arid/semi-arid conditions.

Line 183: “hand harvested manually” remove manually.

Material and methods:

Line 268-271: Please provide models that have been used, including fix and random parts. Moreover, did you check if 1) the responses for each factor level have a normal distribution, 2) If these distributions have the same variance and 3) if the data are independent? Otherwise, you should opt for gl(m)m assuming other data distributions. You just have 3 replications, were they enough for studying the interactions of your experiment? Were the residuals of your models normally distributed? You should add more information for this section. In particular I would suggest the author to include the structure of model that has been used (factor that you used in the analysis and if some factors were nested in others) and the procedure for model selection.

Results:

Line 300-314: I would move this part in MM section.

Line 323: here and throughout the text use the same number of decimals for mean and SE

Line 459-461: please, specify better that you are referring to the early established barley.

Line 461-463: This result is not clearly presented. Please rephase.

Line 468: Express yield as t ha-1 instead of kg ha-1

Line 472-474: This is discussion and not results.

Line 478: what is the biological yield. Add information on the procedure that you used for the determination of biological yield in MM section

Line 478-493: Appropriate annotation should be used. t/ha instead of kg/ha. This section is almost impossible to read. The English should be significantly improved.

Tables:

Table 1: “(mg/kg-1)” this is not the correct annotation. Please use mg/kg or mg kg-1. Moreover, you recorded N tot 0.75 mg/kg and it looks like very low. Check the measure unit, 0.75 g/kg seems more reasonable.

Table 3: what is the difference between “Year” and “replication (year)”

Table 5 and 6: I would remove the capital letters. You mentioned that Year x Irrigation was significant so why did you perform the post-hoc test within years?.

Figures:

I would suggest using some colors for increasing the readability of figures.

Reduce the number of figures. Information included in figures 3,4 and 5 can be presented in just one figure using colors and multiple bars.

Figure 1,2,3,4,5 and 6:

• Bars represent mean±SE instead of “Vertical bars represent ±SE”.

• “In each irrigation regime, means with a letter in common should not be considered different according to the Tukey's test at p≤0.05” this is not clear. I suggest to rephase as: Values with different letters are not significantly different at 0.05 confidence level.

Figure 3 and 4: Use t/ha instead of kg/ha

7. PLOS authors have the option to publish the peer review history of their article (what does this mean?). If published, this will include your full peer review and any attached files.

Reviewer #2: No

---

## [Author Response · Author response to Decision Letter 1]

29 Apr 2023

Dear Dr. Sergio Saia

Editor of Plos One

I sent the tracked-change revision of our manuscript entitled ''Influence of relay intercropping of barley with chickpea on biochemical characteristics and yield under water stress'' (PONE-D-22-21858) as well as Response to Reviewers by attachments. Also, I replied to reviewer's comments as below:

Additional Editor Comments:

Deal authors, one of the 2 former reviewers accepted to re-review your ms. I frankly have to say that I almost agree with the reviewer. In particular, the english language can be improved by yourself trying to use simpler sentences and making a general polishing of the work. So far, my greatest worry is about the statistical analysis and the reviewer pointed to the need of ensuring its reliability, by suggesting the GLMM (which, by the way, can also handle non normal distributed data o residues and few replications). I please exhort you to carefully take into account that the statistical analysis should be done reducing the false positive and false negative results, with a certain degree of protection of the results given by proper statistics (e.g. Benjamini and Hochberg or Tukey-Kramer, etc.)

Author response: The model used for variables in analysis of variance was fix model for all variables including year, irrigation regime and cropping system. In order to check the normality distribution of data, Kolmogorov-Smirnov and Shapori-Wilk tests were used and the skewness and kurtosis indices of data proved that the distribution of data was normal. F-test was carried out to check the equality of variances. The residuals of model were also normal using q-q plot. By the way, most of the researchers use 3 replicates and is often enough for such field experiments. Data was analyzed by SAS software 2012 (version 9.4) and the means were compared by Tukey-Kramer test at 0.05 probability level (p≤ 0.05). 

Also, The English language of the manuscript has been improved, again. See the text, please. 

Regarding the statistics, please pass over to the indication of the rev. regarding "Values with different letters are not significantly different at 0.05 confidence level." The correct statement is "Values with a letter in common should not be considered different at 0.05 probability.", which poses an additional degree of protection of the results.

Author response: It's done in the text.

Is the manuscript technically sound, and do the data support the conclusions?

Reviewer #2: Partly

3. Has the statistical analysis been performed appropriately and rigorously?

Reviewer #2: No

Author response: See the text, please.

4. Have the authors made all data underlying the findings in their manuscript fully available?

Reviewer #2: Yes

5. Is the manuscript presented in an intelligible fashion and written in standard English?

Reviewer #2: No

Author response: The English language of the manuscript has been improved, again. See the text, please. 

6. Review Comments to the Author

Reviewer #2: The authors mentioned that the English has been revised in the manuscript but it is seems not: the paper is difficult to read, especially for the results section. Moreover, there are too many tables and figures. I would suggest removing some tables or move them in supplementary material and to add relevant information in the main text. Figures 3, 4 and 5 for instance can be merged in 1 graph. I still have some concerns about statistical analysis (see specific comments below). I’m convinced that your data and research topic is interesting however, I would suggest the authors for an extensive English check otherwise the MS could be not considered for the publication.

See specific comments below.

Author response: The text has been revised again, especially results section. Also, according to your previous comment, and because of the significant effect of the year × irrigation regime × cropping system on considered traits, the data of two years for barley and chickpea were presented, separately in tables and figures. 

Abstract:

Line 25: “as split plot based on a randomized complete block design” I would remove this information from the abstract.

Author response: It's removed from the abstract

Line 37-30: I would suggest removing this detailed sentence from the abstract, it is more appropriate for materials and methods section. It is enough informative if you mention that the second factor was the sowing time (December vs January).

Author response: Cropping systems as sub plot consisted of sole and relay intercropping of barley with chickpea in two sowing dates (December vs January).

Line 31: “Chlorophyll a content of barley increased in b1c2, by consuming less water compared to sole cropping.” Better if you explain it in terms of…In case of relay intercropping, the early establishment of barley determined an higher chlorophyll content in biomass compared to… due to the… Please try to be more precise in your statement, why b1c1 consumed less water? Less competition from chickpea? Early establishment ensured a growth advantage and therefore a better use of water?

Line 33: What do you mean with “(b1c1 and b1c2) created a suitable canopy to pigment contents improvement”.

Author response: Under drought, the early establishment of barley in December intercropped with chickpea in January (b1c2) enhanced the chlorophyll content by 16% compared to sole cropping due to the less competition with chickpea.

Line 34 “carotenoid content of chickpea” please specify if you are referring to chickpea biomass or grains.

Author response: Late sowing of chickpea enhanced the leaf carotenoid content of chickpea, catalase and peroxidase activities.

Line 35-36 “Barley-chickpea intercropping reacted to water deficit through enzymes activity, water use efficiency and land equivalent enhancement compared to sole cropping”. The WUE and LER are not “reactions”. It is better saying that intercropping enhanced the WUE and guaranteed a more efficient use of space (LER>1) compared with sole crops.

Author response: Barley-chickpea intercropping enhanced the WUE and guaranteed a more efficient use of space (land equivalent ratio of more than 1) compared with sole crops.

Line 36-38 “Under water stress, in b1c2, by increasing total chlorophyll and water use efficiency, grain yield of barley enhanced compared to b1”. This sentence is not clear at all.

Author response: Under water stress, in b1c2 enhancement of total chlorophyll and water use efficiency caused to increase the grain yield of barley.

Line 38: “It seems”. Not appropriate for abstract

Author response: The barley and chickpea in b1c2, responded to water stress by increasing total chlorophyll and enzymes activity, respectively.

Introduction:

My impression is that sentences are not linked to each other. The introduction needs to be rephased in order to improve the readability of the paper. Moreover, make clear your objectives and hypothesis. I strongly suggest submitting the manuscript for an extensive English revision by experts.

Author response: In introduction section, first the intercropping and relay intercropping systems were defined and then the phrases and topics were linked as below: 

The importance of relay intercropping of legumes with cereals, the importance of barley and chickpea in Iran, the role water stress in the late season on crop production in semi-arid areas, the role of water stress in decreasing the photosynthetic pigments and RWC, and increasing enzyme activities. Finally, we describe the problems of crop production in Iran due to water stress and describe the hypothesis and objectives. 

Line 104: carried out instead of “laid out”

Author response: It's done in the text.

Line 127: Is it Darab cv a registered cultivar of chickpea or an ecotype?

Author response: Darab cv is a registered cultivar.

Line 132: “which is suitable for hot and dry areas” better say suitable for arid/semi-arid conditions.

Author response: It's done in the text.

Line 183: “hand harvested manually” remove manually.

Author response: It's done in the text.

Material and methods:

Line 268-271: Please provide models that have been used, including fix and random parts. Moreover, did you check if 1) the responses for each factor level have a normal distribution, 2) If these distributions have the same variance and 3) if the data are independent? Otherwise, you should opt for gl(m)m assuming other data distributions. You just have 3 replications, were they enough for studying the interactions of your experiment? Were the residuals of your models normally distributed? You should add more information for this section. In particular I would suggest the author to include the structure of model that has been used (factor that you used in the analysis and if some factors were nested in others) and the procedure for model selection.

Author response: The model used for variables in analysis of variance was fix model for all variables including year, irrigation regime and cropping system. In order to check the normality distribution of data, Kolmogorov-Smirnov and Shapori-Wilk tests were used and the skewness and kurtosis indices of data proved that the distribution of data was normal. The F-test was carried out to check the equality of variances. The residuals of the model were also normal using q-q plot. By the way, most of the researchers use 3 replicates and is often enough for such field experiments. Data was analyzed by SAS software 2012 (version 9.4) and the means were compared by Tukey-Kramer test at 0.05 probability level (p≤ 0.05). Because of significant effect of year × irrigation regime × cropping system on considered traits, the data of two years for barley and chickpea presented, separately. 

Results:

Line 300-314: I would move this part in MM section.

Author response: This part's moved to MM section.

Line 323: here and throughout the text use the same number of decimals for mean and SE.

Author response: It's done in the text.

Line 459-461: please, specify better that you are referring to the early established barley Author response: The early establishment of barley in December intercropped with late sowing of chickpea in January improved the competition ability of barley through the faster growth. 

Line 461-463: This result is not clearly presented. Please rephase.

Author response: Under water stress in b1c2 treatment grain yield of barley was enhanced 19 and 35% compared to b1 in 2017 and 2018, respectively.

Line 468: Express yield as t ha-1 instead of kg ha-1.

Author response: It's done in the text and Figures.

Line 472-474: This is discussion and not results. 

Author response: It's moved to discussion section.

Line 478: what is the biological yield. Add information on the procedure that you used for the determination of biological yield in MM section.

Author response: At the crop maturity stage on May, the plants in the central 1 m2 of each plot were hand harvested. Then, the samples were oven dried at 72 °C for 48 h and weighted for biological yield.

Line 478-493: Appropriate annotation should be used. t/ha instead of kg/ha. This section is almost impossible to read. The English should be significantly improved.

Author response: This section was revised. 

Tables:

Table 1: “(mg/kg-1)” this is not the correct annotation. Please use mg/kg or mg kg-1. Moreover, you recorded N tot 0.75 mg/kg and it looks like very low. Check the measure unit, 0.75 g/kg seems more reasonable.

Author response: The unit's changed to mg/kg in table 1, but 0.75 mg/kg is correct according to soil test.

Table 3: what is the difference between “Year” and “replication (year)”

Author response: This is Replication × Year interaction.

Table 5 and 6: I would remove the capital letters. You mentioned that Year x Irrigation was significant so why did you perform the post-hoc test within years?

Author response: I think the previous comment of Reviewer #2 about the Tables was as: 

'' Table 5: You mentioned that “year” and “irrigation regime” are significant factors. In this case you should perform the statistical analysis separately for each year and for each irrigation regime. In the table you have assigned letters of significance in the wrong way. Please perform a post hoc separately for year and irrigation regime and eventually assign capital letter for indicating differences between years. You should include standard error (SE).

Table 6: see comments above

Table 7: see comments above

Table 8: see comments above''

Therefore, because of the previous comment of Reviewer #2 in all of the tables, each irrigation regime and year were analyzed, separately and two years assign with capital letter for indicating differences between years. 

I would suggest using some colors for increasing the readability of figures.

Reduce the number of figures. Information included in figures 3,4 and 5 can be presented in just one figure using colors and multiple bars.

Figure 1,2,3,4,5 and 6:

• Bars represent mean±SE instead of “Vertical bars represent ±SE”.

Author response: I think the previous comment of Reviewer #2 about the Figures was as: ''Figure 1: You should perform post-hoc of your model separately for 2017 and 2018. Use lowercase letter for 2017 and uppercase letter for 2018. SE are missing, please provide them. In case that year is not a significant factor you could present data as average between 2017 and 2018 for each of the treatments. Another option can be present data on 2017 and 2018 in two separate graphs''. Same for figure 2, 3, 4, 5, 6, 7.

Therefore, because of the previous comment of Reviewer #2 and significant effect of year on all of the traits and according to previous comment of Reviewer #2 in figure 1 and the other figures, data for each year were presented, separately.

 “In each irrigation regime, means with a letter in common should not be considered different according to the Tukey's test at p≤0.05” this is not clear. I suggest to rephase as: Values with different letters are not significantly different at 0.05 confidence level.

Figure 3 and 4: Use t/ha instead of kg/ha

Author response: It's done in the text and Figures according to Editor's comment.

Sincerely,

Corresponding author

---

## [Editor Report · Decision Letter 2]

25 May 2023

Influence of relay intercropping of barley with chickpea on biochemical characteristics and yield under water stress

PONE-D-22-21858R2

Dear Dr. Bijanzadeh,

We’re pleased to inform you that your manuscript has been judged scientifically suitable for publication and will be formally accepted for publication once it meets all outstanding technical requirements.

Kind regards,

Sergio Saia, Ph.D.

Academic Editor

PLOS ONE

Additional Editor Comments (optional):

in the galley proof, please correct the word "Shapori" to "Shapiro"
---

## [Editor Report · Acceptance letter]

1 Jun 2023

PONE-D-22-21858R2 

Influence of relay intercropping of barley with chickpea on biochemical characteristics and yield under water stress 

Dear Dr. Bijanzadeh:

I'm pleased to inform you that your manuscript has been deemed suitable for publication in PLOS ONE. Congratulations! Your manuscript is now with our production department. 

Kind regards, 

on behalf of

prof Sergio Saia 

Academic Editor

PLOS ONE